# GLOBAL OPTIMALITY OF SOFTMAX POLICY GRADIENT WITH SINGLE HIDDEN LAYER NEURAL NETWORKS IN THE MEAN-FIELD REGIME

**Andrea Agazzi,**
Department of Mathematics,
Duke University,
`agazzi@math.duke.edu`

**Jianfeng Lu**
Department of Mathematics,
Department of Physics and Department of Chemistry,
Duke University
`jianfeng@math.duke.edu`

## ABSTRACT

We study the problem of policy optimization for infinite-horizon discounted Markov Decision Processes with softmax policy and nonlinear function approximation trained with policy gradient algorithms. We concentrate on the training dynamics in the *mean-field* regime, modeling *e.g.*, the behavior of wide single hidden layer neural networks, when exploration is encouraged through entropy regularization. The dynamics of these models is established as a Wasserstein gradient flow of distributions in parameter space. We further prove global optimality of the fixed points of this dynamics under mild conditions on their initialization.

## 1 INTRODUCTION

In recent years, deep reinforcement learning has revolutionized the world of Artificial Intelligence by outperforming humans in a multitude of highly complex tasks and achieving breakthroughs that were deemed unthinkable at least for the next decade. Spectacular examples of such revolutionary potential have appeared over the last few years, with reinforcement learning algorithms mastering games and tasks of increasing complexity, from learning to walk to the games of Go and Starcraft (Mnih et al., 2013; 2015; Silver et al., 2016; 2017; 2018; Haarnoja et al., 2018a; Vinyals et al., 2019).

In most cases, the main workhorse allowing artificial intelligence to pass such unprecedented milestones was a variation of a fundamental method to train reinforcement learning models: *policy gradient* (PG) algorithms (Sutton et al., 2000). This algorithm has a disarmingly simple approach to the optimization problem at hand: given a parametrization of the policy, it updates the parameters in the direction of steepest ascent of the associated integrated value function. Impressive progress has been made recently in the understanding of the convergence and optimization properties of this class of algorithms in the tabular setting (Agarwal et al., 2019; Cen et al., 2020; Bhandari & Russo, 2019), in particular leveraging the natural tradeoff between exploration and exploitation offered for entropy-regularized rewards by softmax policies (Haarnoja et al., 2018b; Mei et al., 2020). However, this simple algorithm alone is not sufficient to explain the multitude of recent breakthroughs in this field: in application domains such as Starcraft, robotics or movement planning, the space of possible states and actions are exceedingly large – or even continuous – and can therefore not be represented efficiently by tabular policies (Haarnoja et al., 2018a). Consequently, the recent impressive successes of artificial intelligence would be impossible without the natural choice of neural networks to approximate value functions and / or policy functions in reinforcement learning algorithms (Mnih et al., 2015; Sutton et al., 2000).

While neural networks, in particular deep neural networks, provide a powerful and versatile tool to approximate high dimensional functions on continuous spaces (Cybenko, 1989; Hornik, 1991; Barron, 1993), their intrinsic nonlinearity poses significant obstacles to the theoretical understanding of their training and optimization properties. For instance, it is known that the optimization landscape of these models is highly nonconvex, preventing the use of most theoretical tools from classical optimization theory. For this reason, the unprecedented success of neural networks in artificial intelligence stands in contrast with the poor understanding of these methods from a theoretical perspective. Indeed, even in the supervised setting, which can be viewed as a special case of reinforcement learning, deep neural networks are still far from being understood despite having been an important and fashionable research focus in recent years. Only recently, a theory of neural network learning has started to emerge, including recent works on mean-field point of view of training dynamics (Mei et al.,

2018; Rotskoff & Vanden-Eijnden, 2018; Rotskoff et al., 2019; Wei et al., 2018; Chizat & Bach, 2018) and on linearized dynamics in the over-parametrized regime (Jacot et al., 2018; Allen-Zhu et al., 2018; Du et al., 2018; 2019; Zou et al., 2018; Allen-Zhu et al., 2019; Chizat et al., 2019; Oymak & Soltanolkotabi, 2020; Ghorbani et al., 2019; Lee et al., 2019). More specifically to the context of reinforcement learning, some works focusing on value-based learning (Agazzi & Lu, 2019; Cai et al., 2019; Zhang et al., 2020), and others exploring the dynamics of policy gradient algorithms (Zhang et al., 2019) have recently appeared. Despite this progress, the theoretical understanding of deep reinforcement learning still poses a significant challenge to the theoretical machine learning community, and it is of crucial importance to understand the convergence and optimization properties of such algorithms to bridge the gap between theory and practice.

CONTRIBUTIONS.

The main goal of this work is to investigate entropy-regularized policy gradient dynamics for wide, single hidden layer neural networks. In particular, we give the following contributions:

- We give a mean-field formulation of policy gradient dynamics in parameter space, describing the evolution of neural network parameters in the form of a transport partial differential equation (PDE). We prove convergence of the particle dynamics to their mean-field counterpart. We further explore the structure of this problem by showing that such PDE is a gradient flow in the Wasserstein space for the appropriate energy functional.

- We investigate the convergence properties of the above dynamics in the space of measures. In particular, we prove that under some mild assumptions on the initialization of the neural network parameters and on the approximating power of the nonlinearity, all fixed points of the dynamics are global optima, *i.e.*, the approximate policy learned by the neural network is optimal,

RELATED WORKS.

Recent progress in the understanding of the parametric dynamics of simple neural networks trained with gradient descent in the supervised setting has been made in (Mei et al., 2018; Rotskoff & Vanden-Eijnden, 2018; Wei et al., 2018; Chizat, 2019; Chizat & Bach, 2020). These results have further been extended to the multilayer setting in (Nguyen & Pham, 2020). In particular, the paper (Chizat & Bach, 2018) proves optimality of fixed points for wide single layer neural networks leveraging a Wasserstein gradient flow structure and the strong convexity of the loss functional WRT the predictor. We extend these results to the reinforcement learning framework, where the convexity that is heavily leveraged in (Chizat & Bach, 2018) is lost. We bypass this issue by requiring a sufficient expressivity of the used nonlinear representation, allowing to characterize global minimizer as optimal approximators.

The convergence and optimality of policy gradient algorithms (including in the entropy-regularized setting) is investigated in the recent papers (Bhandari & Russo, 2019; Mei et al., 2020; Cen et al., 2020; Agarwal et al., 2019). These references establish convergence estimates through gradient domination bounds. In (Mei et al., 2020; Cen et al., 2020) such results are limited to the tabular case, while (Agarwal et al., 2019; 2020) also discuss neural softmax policy classes, but under a different algorithmic update and assuming certain well-conditioning assumptions along training. Furthermore, all these results heavily leverage the finiteness of action space. In contrast, this paper focuses on the continuous space and action setting with nonlinear function approximation.

Further recent works discussing convergence properties of reinforcement learning algorithms with function approximation via neural networks include (Zhang et al., 2019; Cai et al., 2019). These results only hold for finite action spaces, and are obtained in the regime where the network behaves essentially like a linear model (known as the neural or lazy training regime), in contrast to the results of this paper, which considers training in a nonlinear regime. We also note the work (Wang et al., 2019) where the action space is continuous but the training is again in an approximately linear regime.

## 2 MARKOV DECISION PROCESSES AND POLICY GRADIENTS

We denote a Markov Decision Process (MDP) by the 5-tuple $(\mathcal{S}, \mathcal{A}, P, r, \gamma)$, where $\mathcal{S}$ is the state space, $\mathcal{A}$ is the action space, $P = P(s, a, s')_{s,s' \in \mathcal{S}, a \in \mathcal{A}}$ a Markov transition kernel, $r(s, a, s')_{s,s' \in \mathcal{S}, a \in \mathcal{A}}$ is the real-valued, bounded and continuous immediate reward function and $\gamma \in (0, 1)$ is a discount factor. We will consider a probabilistic *policy*, mapping a state to a probability distribution on the action space, so that $\pi : \mathcal{S} \to \mathcal{M}_+^1(\mathcal{A})$, where $\mathcal{M}_+^1(\mathcal{A})$ denotes the space of probability measures on $\mathcal{A}$, and denote for any $s \in \mathcal{S}$ the corresponding density $\pi(s, \cdot) : \mathcal{A} \to \mathbb{R}_+$. The policy defines a state-to-state transition operator

$P_\pi(s, \mathrm{d}s') = \int_{\mathcal{A}} P(s, a, \mathrm{d}s') \pi(s, \mathrm{d}a)$, and we assume that $P_\pi$ is Lipschitz continuous as an operator $M_+^1(\mathcal{S}) \to M_+^1(\mathcal{S})$ wrt the policy. We further encourage exploration by defining some (relative) *entropy-regularized* rewards (Williams & Peng, 1991)

$$R_\tau(s, a, s') = r(s, a, s') - \tau D_{\mathrm{KL}}(\pi(s, \cdot); \bar{\pi}(\cdot)),$$

where $D_{\mathrm{KL}}$ denotes the relative entropy, $\bar{\pi}$ is a reference measure and $\tau$ indicates the strength of regularization. Throughout, we choose $\bar{\pi}$ to be the Lebesgue measure on $\mathcal{A}$, which we assume, like $\mathcal{S}$, to be a compact subset of the Euclidean space. This regularization encourages exploration and absolute continuity of the policy WRT Lebesgue measure. Consequently, with some abuse of notation, we use throughout the same notation for a distribution and its density in phase space. Note that the original, unregularized MDP can be recovered in the limit $\tau \to 0$.

In this context, given a policy $\pi$ the associated *value function* $V_\pi : \mathcal{S} \to \mathbb{R}$ maps each state to the infinite-horizon expected discounted reward obtained by following the policy $\pi$ and the Markov process defined by $P$:

$$V_\pi(s) = \mathbb{E}_\pi \left[ \sum_{t=0}^\infty \gamma^t R_\tau(s_t, a_t, s_{t+1}) \Big| s_0 = s \right] \tag{1}$$

$$= \mathbb{E}_\pi \left[ \sum_{t=0}^\infty \gamma^t \big( r(s_t, a_t, s_{t+1}) - \tau D_{\mathrm{KL}}(\pi(s_t, \cdot); \bar{\pi}(\cdot)) \big) \Big| s_0 = s \right],$$

where $\mathbb{E}_\pi[\cdot | s_0 = s]$ denotes the expectation of the stochastic process $s_t$ starting at $s_0 = s$ and following the (stochastic) dynamics defined recursively by the transition operator $P_\pi(s, \mathrm{d}s') = \int P(s, a, \mathrm{d}s') \pi(s, \mathrm{d}a)$. Correspondingly, we define the $Q$-function $Q_\pi : \mathcal{S} \times \mathcal{A} \to \mathbb{R}$ as

$$Q_\pi(s, a) = \mathbb{E}_\pi \left[ r(s_0, a_0, s_1) + \sum_{t=1}^\infty \gamma^t R_\tau(s_t, a_t, s_{t+1}) \Big| s_0 = s, a_0 = a \right]$$

$$= \bar{r}(s, a) + \gamma \mathbb{E}_\pi \left[ V_\pi(s_1) \Big| s_0 = s, a_0 = a \right], \tag{2}$$

where $\bar{r}(s, a) = \mathbb{E}[r(s, a, s')]$ is the average reward from $(s, a)$. Conversely, from the definition, we have the identity for $V_\pi(s) = \mathbb{E}_\pi[Q_\pi(s_0, a_0)|s_0 = s] - \tau D_{\mathrm{KL}}(\pi(s, \cdot); \bar{\pi}(\cdot))$.

We are interested in learning the optimal policy $\pi^*$ of a given MDP $(\mathcal{S}, \mathcal{A}, P, r, \gamma)$, which satisfies for all $s \in \mathcal{S}$

$$V_{\pi^*}(s) = \max_{\pi : \mathcal{S} \to \mathcal{M}_+^1(\mathcal{A})} V_\pi(s). \tag{3}$$

More specifically we would like to estimate this function through a family of approximators $\pi_w : \mathcal{S} \to \mathcal{M}_+^1(\mathcal{A})$ parametrized by a vector $w \in \mathcal{W} := \mathbb{R}^p$. Note that since we consider entropy-regularized rewards, the optimal policy will be a probabilistic policy (given as a Boltzmann distribution) instead of a deterministic one.

A popular algorithm to solve this problem is given by *policy gradient* algorithm (Sutton & Barto, 2018). Starting from an initial condition $w(0) \in \mathcal{W}$, this algorithm updates the parameters $w$ of the predictor in the direction of steepest ascent of the average reward

$$w(t+1) := w(t) + \beta_t \nabla_w \tilde{\mathbb{E}}_{s \sim \varrho_0} V_\pi(s), \tag{4}$$

for a *fixed* absolutely continuous initial distribution of initial states $\varrho_0 \in \mathcal{M}_+^1(\mathcal{S})$ and sequence of time steps $\{\beta_t\}_t$. Here $\tilde{\mathbb{E}}[\cdot]$ denotes an approximation of the expected value operator.

This work investigates the regime of asymptotically small constant step-sizes $\beta_t \to 0$. In this adiabatic limit, the stochastic component of the dynamics is averaged before the parameters of the model can change significantly. This allows to consider the parametric update as a deterministic dynamical system emerging from the averaging of the underlying stochastic algorithm corresponding to the limit of infinite sample sizes. This is known as the ODE method (Borkar, 2009) for analyzing stochastic approximation. We focus on the analysis of this deterministic system to highlight the core dynamical properties of policy gradients with nonlinear function approximation. The averaged, deterministic dynamics is given by the set of ODEs

$$\frac{\mathrm{d}}{\mathrm{d}t} w(t) = \mathbb{E}_{s \sim \varrho_0} \left[ \nabla_w V_\pi(s) \right] = \mathbb{E}_{s \sim \varrho_\pi, a \sim \pi_w} \left[ \nabla_w \log \pi_w(s, a) \left( Q_\pi(s, a) - \tau \log(\pi_w(s, a)) \right) \right], \tag{5}$$

where in the second equality we have applied the policy gradient theorem (Sutton et al., 2000; Sutton & Barto, 2018), defining for a fixed $\varrho_0 \in \mathcal{M}^1_+(\mathcal{S})$

$$\varrho^\pi(s_0, s) := \sum_{t=0}^{\infty} \gamma^t P^t_\pi(s_0, s), \qquad \varrho_\pi(s) = \int_{\mathcal{S}} \varrho^\pi(s_0, s)\varrho_0(\mathrm{d}s_0), \tag{6}$$

as the (improper) discounted empirical measure. For completeness, we include a derivation of (5) in Appendix A.

SOFTMAX POLICIES IN THE MEAN-FIELD REGIME

We choose to represent our policy as a softmax policy:

$$\pi_w(s, a) = \frac{\exp(f_w(s, a))}{\int_{\mathcal{A}} \exp(f_w(s, a))\mathrm{d}a}$$

and parametrize the *energy* $f$ as a two-layer neural network in the *mean-field* regime, *i.e.*,

$$f_w(s, a) = \frac{1}{N} \sum_{i=1}^{N} \psi(s, a; w_i)$$

for a fixed, usually nonlinear function $\psi : \mathcal{S} \times \mathcal{A} \times \Omega \to \mathbb{R}$, where we have separated $w \in \mathcal{W}$ into $N$ identical components $w_i \in \Omega$ so that $\mathcal{W} = \Omega^N$.

We can rewrite the above expression in terms of an empirical measure:

$$f_{\nu^{(N)}}(s, a) := \int_{\Omega} \psi(s, a; \omega)\nu^{(N)}(\mathrm{d}\omega) \qquad \text{where} \quad \nu^{(N)}(\mathrm{d}\omega) = \frac{1}{N} \sum_{i=1}^{N} \delta_{w^{(i)}}(\mathrm{d}\omega) \in \mathcal{M}^1_+(\Omega). \tag{7}$$

This empirical measure representation removes the symmetry of the approximating functions under permutations of parameters $w_i$. It also facilitates the limit $N \to \infty$, when $\nu^{(N)} \to \nu$ weakly, so that $f_{\nu^{(N)}} \to f_\nu$. Then, for a general distribution $\nu \in \mathcal{M}^1_+(\Omega)$ the softmax mean-field policy reads:

$$\pi_\nu(s, a) = \frac{\exp\left(\int_{\Omega} \psi(s, a; \omega)\nu(\mathrm{d}\omega)\right)}{\int_{\mathcal{A}} \exp\left(\int_{\Omega} \psi(s, a; \omega)\nu(\mathrm{d}\omega)\right)\mathrm{d}a}. \tag{8}$$

Note that by our choice of softmax policy and mean-field parametrization (7) we have

$$\nabla_{w_i} \log \pi_{\nu^{(N)}}(s, a) = \nabla_{w_i} f_{\nu^{(N)}}(s, a) - \nabla_{w_i} \log \int_{\mathcal{A}} \exp f_{\nu^{(N)}}(s, a)\mathrm{d}a$$

$$= \nabla_{w_i} \frac{1}{N} \sum_{i=1}^{N} \psi(s, a; w_i) - \frac{\int_{\mathcal{A}} \nabla_{w_i} \exp\left[\frac{1}{N} \sum_{i=1}^{N} \psi(s, a; w_i)\right] \mathrm{d}a}{\int_{\mathcal{A}} \exp f_{\nu^{(N)}}(s, a)\mathrm{d}a}$$

$$= \frac{1}{N} \left(\nabla_{w_i} \psi(s, a; w_i) - \int_{\mathcal{A}} \nabla_{w_i} \psi(s, a; w_i)\pi_{\nu^{(N)}}(s, \mathrm{d}a)\right).$$

Thus the training dynamics (5), after an appropriate rescaling of time ($t \mapsto t/N$, which is due to the mean-field parametrization for $f_w$), can be rewritten as

$$\frac{\mathrm{d}}{\mathrm{d}t} w_i(t) = \int_{\mathcal{S} \times \mathcal{A}} \left(\nabla_{w_i} \psi(s, a; w_i) - \mathbb{E}_{\pi_{\nu^{(N)}}} [\nabla_{w_i} \psi(s, \cdot; w_i)]\right) \times$$

$$\times \left(Q_{\pi_{\nu^{(N)}}}(s, a) - \tau \log \pi_{\nu^{(N)}}(s, a)\right)\pi_{\nu^{(N)}}(s, \mathrm{d}a)\varrho_{\pi_{\nu^{(N)}}}(\mathrm{d}s). \tag{9}$$

The training dynamics can be more compactly represented by the evolution of the measure $\nu \in \mathcal{M}^1_+(\Omega)$ in parameter space, given by a *mean-field* transport partial differential equation of the Vlasov type as

$$\frac{\mathrm{d}}{\mathrm{d}t} \nu_t(\omega) = \mathrm{div} \left(\nu_t(\omega) \int_{\mathcal{S} \times \mathcal{A}} C_{\pi_\nu} [\nabla_\omega \psi(s, \cdot; \omega), Q_{\pi_\nu} - \tau \log \pi_\nu](s)\varrho_{\pi_\nu}(\mathrm{d}s)\right), \tag{10}$$

where $\omega \in \Omega$ and we have introduced the shorthand $C_\pi[f, g](s)$ to denote the covariance operator WRT the probability measure $\pi(s, \mathrm{d}a)$. Note that the above partial differential equation also captures the dynamics of the finite-width system, *i.e.*, of the empirical measure $\nu^{(N)}$ where each $w_i$ follows (9).

We further note that the dynamics introduced above have a gradient flow structure in the probability space $\mathcal{M}_+^1(\Omega)$: defining the expected value function

$$\mathcal{E}[\nu] = \mathbb{E}_{s_0 \sim \varrho_0}\left[V_{\pi_\nu}(s_0)\right] \tag{11}$$

the dynamics (10) is a gradient flow for $\mathcal{E}$ in the Wasserstein space (see e.g., (Santambrogio, 2017) for an introduction), as we prove in the appendix:

**Proposition 2.1.** *For a fixed initial distribution $\varrho_0 \in \mathcal{M}_+^1(\mathcal{S})$, the dynamics (10) is the Wasserstein gradient flow of the energy functional (11).*

Analogous dynamics equation for evolution of parameter space measure in the supervised learning case has been derived in (Mei et al., 2018; Rotskoff & Vanden-Eijnden, 2018; Chizat & Bach, 2018) and in the TD learning case in (Agazzi & Lu, 2019; Zhang et al., 2020). In particular, in the case of supervised learning, the resulting dynamics is a Wasserstein gradient flow, the structure of which is used to obtain the convergence of the particle system to the mean-field dynamics. In our case, however, the energy functional is not convex WRT the policy and moreover the softmax parametrization destroys the convexity of the approximator of the policy with respect to $\nu_t$. Thus showing convergence of the dynamics becomes much more challenging.

## 3 SIMPLIFIED SETTING: THE BANDIT PROBLEM

We now introduce our results in the simple bandit setting, where state space $\mathcal{S}$ is one point (and will be henceforth suppressed in the notation) and without loss of generality action space $\mathcal{A}$ is continuous. In this case, for a reward function $r$ and a softmax policy

$$\pi_\nu(a) = \frac{\exp(f_\nu(a))}{\int_{\mathcal{A}} \exp(f_\nu(a))\mathrm{d}a}$$

we have that the value function for the regularized problem reads (we denote $V_\nu = V_{\pi_\nu}$ to simplify notation)

$$V_\nu = \int (r(a) - \tau \log \pi_\nu(a))\pi_\nu(\mathrm{d}a),$$

while the $Q$ function is simply $Q(a) = r(a)$. We further note that the optimal policy in the regularized case reads:

$$\pi^*(a) = Z^{-1}\exp(\tau^{-1}r(a)), \qquad Z = \int_{\mathcal{A}}\exp(\tau^{-1}r(a))\mathrm{d}a$$

Recalling the definition of the covariance operator $C_\pi[f, g](s)$ from (10), the expression for the policy gradient vector field in the latter case simplifies to

$$\partial_t\omega_t := F_t(\omega_t; \nu_t) = \nabla_\omega D\pi_\nu DV_\nu = C_{\pi_\nu}[\nabla_\omega\psi(a; \omega), r - \tau\log(\pi_\nu)]$$
$$= \int_{\mathcal{A}}\left(\nabla_\omega\psi(a; \omega) - \int \nabla_\omega\psi(a'; \omega)\pi_\nu(\mathrm{d}a')\right)(r(a) - \tau f_\nu(a))\,\pi_\nu(\mathrm{d}a), \tag{12}$$

where $D\pi_\nu, DV_\nu$ denote the Fréchet derivative of $\pi_\nu$ and $V_\nu$ WRT $\nu$ and $\pi$ respectively. Note that by the structure of the covariance operator $C_\pi$, adding a constant to the function $f_\nu(\cdot)$ does not affect the dynamics. This reflects the fact that the softmax policy is normalized by definition.

### 3.1 GLOBAL OPTIMALITY OF SOFTMAX POLICY GRADIENT

We now sketch the main steps in proving that the mean-field policy gradient dynamics converge, under appropriate assumptions, to global optimizers. The proof in this simpler setting is much more transparent than the general case to be discussed in the next section, and will provide some intuition for the latter. The first part of the proof concerns the properties of fixed points of the dynamics (10), while the second part concerns the training dynamics.

STATICS

We first informally prove global optimality of any fixed point $\nu^*$ of the transport equation $\frac{\mathrm{d}}{\mathrm{d}t}\nu_t = -\mathrm{div}(\nu_t F_t)$ with $F_t$ from (12) such that

a) $\nu^*$ has full support in $\Omega$,

b) the nonlinearity is 1-homogeneous in the first component of its parameters, *i.e.*, that writing $\omega = (\omega_0, \bar{\omega}) \in \mathbb{R} \times \Theta$ one has $\psi(a; \omega) := \omega_0 \phi(a; \bar{\omega})$ for a regular enough $\phi : \mathcal{A} \times \Theta \to \mathbb{R}$,

c) the span of $\{\phi(a; \bar{\omega})\}_{\bar{\omega} \in \Theta}$ is *dense* in $L^2(\mathcal{A})$,

so that *i.e.*, $\pi_{\nu^*}(a) = \pi^*(a) = Z^{-1} \exp\left[\tau^{-1} r(a)\right]$. Weaker assumptions and the general statement are given in the next section, while the general proof appears in the appendix.

First, we note that by assumption a), $\text{div}(\nu^* F(\,\cdot\,; \nu^*)) = 0$ directly implies that for almost all $\omega \in \Omega$

$$F(\omega; \nu^*) = \int_{\mathcal{A}} \nabla_\omega \psi(a; \omega) \left(r(a) - \tau f_{\nu^*}(a) - V_{\nu^*}\right) \pi_{\nu^*}(\mathrm{d}a) = 0\,.$$

In particular, by homogeneity assumption b), the first component of the above vector field must vanish on $\Theta$.

$$\int_{\mathcal{A}} \phi(a; \bar{\omega}) \left(r(a) - \tau f_{\nu^*}(a) - V_{\nu^*}\right) \pi_{\nu^*}(\mathrm{d}a) = 0\,.$$

By assumption c) that span of $\phi$ is dense in $L^2(\mathcal{A})$ the above implies that

$$r(a) - \tau f_{\nu^*}(a) - V_{\nu^*} = 0 \qquad \pi_{\nu^*}\text{-a.e. in } \mathcal{A}\,. \tag{13}$$

Finally, recalling that by the softmax parametrization and by the boundedness of $\phi$, $\pi_{\nu^*}(a) > 0$, we must have

$$f_{\nu^*}(a) = \tau^{-1} r(a) + C$$

which directly implies the optimality of the policy.

### DYNAMICS

While it is clear that assumption b) and c) about the structure and approximating power of the nonlinearity $\psi$ hold independently of $t$, we want to show that assumption a) also holds uniformly in time. In this sense, the continuity of the vector field (12) will preserve the full support properties of the measure $\nu_t$ for all $t > 0$, as we will prove in a more general framework in Lemma C.2. Consequently, any measure $\nu$ respecting assumption a) at initialization will do so for any finite positive time $t > 0$. However, the question remains of whether this property still holds at $t = \infty$. This is the object of Lemma C.3, where we prove that whenever the gradient approaches a fixed point in parameter space, if this fixed point is not a global minimizer, it must be avoided by the dynamics, and thus, the only possible fixed points of the dynamics are global minimizers.

## 4 RESULTS IN THE GENERAL SETTING

We now come back to the general MDP framework introduced in Section 2.

### 4.1 ASSUMPTIONS

To state the main result of this section, the optimality of fixed points of (10) we need the following

**Assumption 1.** *Assume that $\omega = (\omega_0, \bar{\omega}) \in \mathbb{R} \times \Theta$ for $\Theta = \mathbb{R}^{m-1}$ and $\psi(s, a; \omega) = \omega_0 \phi(s, a; \bar{\omega})$ with*

a) Regularity of $\phi$: *$\phi$ is bounded, differentiable and $D\phi_\omega$ is Lipschitz. Also, for all $f \in L^2(\mathcal{S} \times \mathcal{A})$ the regular values of the map $\bar{\omega} \mapsto g_f(\bar{\omega}) := \int f(s, a)\phi(s, a; \bar{\omega})$ are dense in its range, and $g_f(r\bar{\omega})$ converges in $C^1(\{\bar{\omega} \in \Theta : \|\bar{\omega}\|_2 = 1\})$ as $r \to \infty$ to a map $\bar{g}_f(\bar{\omega})$ whose regular values are dense in its range.*

b) Universal approximation: *the span of $\{\phi(\cdot, \bar{\omega}) : \bar{\omega} \in \Theta\}$ is dense in $L^2(\mathcal{S} \times \mathcal{A})$;*

c) Support of the measure: *There exists $r > 0$ s.t. the support of the initial condition $\nu_0$ is contained in $\mathcal{Q}_r := [-r, r] \times \Theta$ and separates $\{-r\} \times \Theta$ from $\{r\} \times \Theta$, i.e., any continuous path connecting $\{-r\} \times \Theta$ to $\{r\} \times \Theta$ intersects the support of $\nu_0$.*

Assumption 1 a) is a common, technical regularity assumption ensuring that (10) is well behaved and controlling the growth, variation and regularity of $\phi$. Alternative assumptions on the case $\Theta \neq \mathbb{R}^{m-1}$ are given in the appendix. Assumption 1 b) speaks to the approximating power of the nonlinearity, assumed to be expressive enough to approximate any function in $L^2(\mathcal{S} \times \mathcal{A})$. This condition replaces the convexity assumption from Chizat & Bach (2018), as the lack of convex structure in our setting prevents us from identifying the local and global minimization properties of a fixed point. Indeed, despite the one-point convexity of $\mathbb{E}_\varrho[V_\pi(s)]$ as a

functional of $\pi$ (Kakade & Langford, 2002) which can be leveraged in the tabular case, such property will be lost, in general, when restricting to policies through nonlinear function approximation. We bypass this issue by requiring sufficient expressivity of the approximating function class, guaranteeing that the optimal policy can be represented with arbitrary precision. Similar assumption on approximability of neural network representation was made in recent analysis of natural policy gradient algorithm (Agarwal et al., 2019). We note that this assumption is easily satisfied by widely used nonlinearities by the universal approximation theorem (Cybenko, 1989; Barron, 1993). Examples of activation functions satisfying Assumption 1 a)-b) include sigmoid, tanh and Gaussian radial function nonlinearities. Extension to analogous results in the ReLU case was discussed in Wojtowytsch (2020) for the supervised learning. Finally, Assumption 1 c) guarantees that the initial condition is such that the expressivity from b) can actually be exploited. This condition is satisfied for example by the product of a uniform distribution on any bounded set $A \subset \mathbb{R}$ with the normal distribution on $\Theta$ or, if $\Theta$ is compact, with the uniform distribution on $\Theta$.

## 4.2 CONVERGENCE OF THE MANY-PARTICLE LIMIT

Before discussing the optimality properties of the dynamics (10), we show that this PDE accurately describes the policy gradient dynamics of a sufficiently wide, single layer neural network. To this aim, we let $\mathcal{P}_2(\Omega)$ be the space of probability distributions on $\Omega$ with finite second moment.

**Theorem 4.1.** *Let Assumption 1 hold and let $w_t^{(N)}$ be a solution of (5) with initial condition $w_0^{(N)} \in \mathcal{W} = \Omega^N$. If $\nu_0^{(N)}$ converges to $\nu_0 \in \mathcal{P}_2(\Omega)$ in Wasserstein distance $W_2$ then $\nu_t^{(N)}$ converges, for every $t > 0$, to the unique solution $\nu_t$ of (10).*

We note that by the law of large numbers for empirical distributions, the condition of convergence of $\nu_0^{(N)}$ to $\nu_0$ is *e.g.*, satisfied when $w_0^{(i)}$ are drawn independently at random from $\nu_0$.

The proof of this result is largely standard, under the given assumptions, and is provided in the appendix for completeness. The idea of the proof is a canonical propagation of chaos argument (Sznitman, 1991). In a nutshell the first step of the argument establishes sufficient regularity of the gradient dynamics, allowing to guarantee existence and uniqueness of the solution to (10). Then, one bounds the difference in differential updates for the particle system and the mean-field dynamics by comparing them with the evolution of the particle system according to a *linear, time-inhomogeneous* PDE using the drift term of the mean-field model. The proof is finally concluded by application of Gronwall inequality. The main difficulty WRT similar results in the literature is to establish the needed Lipschitz continuity of the vector field driving the transport PDE: while this is an immediate consequence of assumptions on the activation functions and on the risk functional in the supervised setting, proving this type of regularity requires more effort in the RL setting given the involved dependence of the vector field on the measure $\nu_t$.

## 4.3 OPTIMALITY

After discussing the connection between particle dynamics and mean-field equations, we present the main convergence result of this paper:

**Theorem 4.2.** *Let Assumption 1 hold and $\nu_t$ given by (10) converge to $\nu^*$, then $\pi_{\nu^*} = \pi^*$, the optimal policy for (3).*

Thus if the policy gradient dynamics (10) converges to a stationary point, that point must be a global minimizer. Again, we emphasize that in our regularized setting $\pi^*$ is given by a probability distribution, and can thus be represented as a softmax policy. We prove this result in three steps. First, we connect the optimality of a stationary point with the support of the underlying measure in parameter space. More specifically, we show in Lemma C.1 that by the expressivity of $\phi$, the transport vector field of suboptimal fixed points of the dynamics (10) cannot vanish everywhere in parameter space. This implies that a measure with sufficient support cannot correspond to a suboptimal fixed point.

We then show in Lemma C.2 that such sufficient notion of support (Assumption 1 c) ) is preserved by the mean-field policy gradient dynamics (10) throughout training. For any finite time, this is true by topological arguments: the separation property of the measure cannot be altered by the action of a continuous vector field such as (10). We note in particular that we do not prove that assumption a) in Section 3 holds in this case.

Finally, in Lemma C.3 we combine the above results and prove that spurious fixed points are avoided by the policy gradient dynamics (10) when initialized properly. To establish this we argue by contradiction: assuming

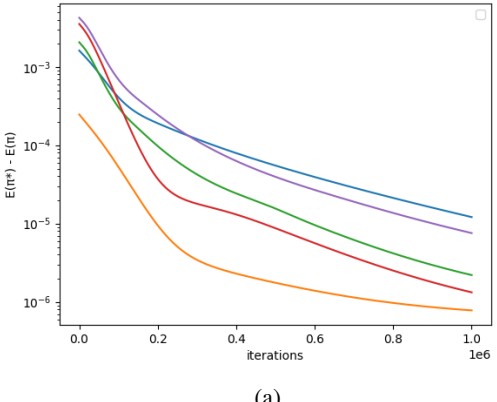 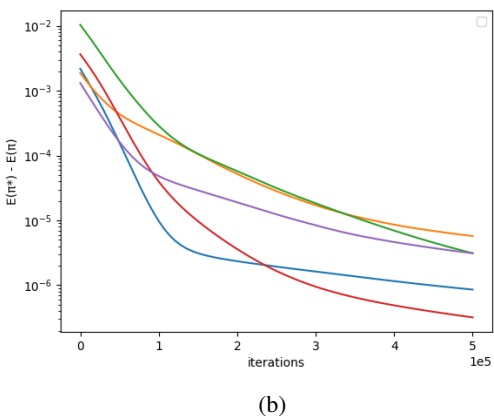

(a)                                        (b)

Figure 1: Evolution of $\mathcal{E}(\nu^*) - \mathcal{E}(\nu_t)$ as a function of training time, for experiments (a) and (b) as described in the main text. Different lines correspond to different random initializations of reward function and learning model. We see that the training error decreases monotonically (but with different rates) during trainig.

that we are approaching such a spurious fixed point $\tilde{\nu}$ at time $t_0$, we show in Lemma C.5 that the velocity field will change little for any $t > t_0$. In particular, it follows that in this regime the dynamics of (10) can be approximated by the gradient descent dynamics (in particle space) of an approximately *fixed* potential. On the other hand, by Assumption 1 c) and by the homogeneity of $\psi$, we are able to show that by Lemma C.2 a positive amount of measure $\tilde{\nu}$ will fall in a forward invariant region where its $\omega_0$ component will grow linearly in $t$ (which exists by Lemma C.1), thereby eventually contradicting the assumption that $\tilde{\nu}$ is a fixed point of (10).

There are two main conceptual differences between the proof outlined above and the one carried out in the supervised learning setting. On one hand, a necessary step in our proof is to establish Lipschitz continuity of the vector field defining the transport equation (10), also needed for convergence of the particle dynamics as discussed above. On the other hand, the landscape of the objective function for policy gradients does not enjoy the convexity (WRT the predictor) typically assumed in the supervised case. To exclude the existence of local minima we assume sufficient expressivity of the activation functions, absent in the supervised analysis. This assumption is key to deduce optimality of fixed points of (10) in Theorem 4.2 in our less regular setting.

## 5   NUMERICAL EXAMPLES

To test our theoretical results in a simple setting we train a wide, single hidden layer neural network with policy gradients to learn the optimal softmax policy (8) for entropy-regularized rewards with parametrization (7) and regularization parameter $\tau = 0.2$. We do so in two separate settings:

(a) $\mathcal{S} = \{0\}$, $\mathcal{A} = [0, 1]$. This setting corresponds to bandits framework discussed in Section 3.

(b) $\mathcal{S} \times \mathcal{A}$ is a grid of size $100 \times 100$ in the set $[0,1]^2$. In this case, we have chosen a discount factor of $\gamma = 0.7$ and a transition process given by $P(s, a, s') = 0.9\delta(s' - a) + 0.1/100$ (*i.e.*, an action $a$ leads to the corresponding state $s' = a$ with probability 0.9 and is uniformly distributed with probability 0.1). At each iteration we have computed the *exact* distribution $\varrho_\pi$ by computing the resolvent of the (weighted) transition matrix.

In both cases, we defined the optimal Q function as $Q^*(s, a) = \tau f_w^*(s, a)$, where $f_w^*(s, a)$ is given by a single hidden layer neural network of width $n = 5$, ReLU nonlinearites and weights $w$ drawn independently and identically distributed from a centered, normal distribution with variance $\sigma^2 = 4$, *i.e.*,

$$Q^*(s, a) = \tau f_w^*(s, a) \qquad \text{for } w_i \sim \mathcal{N}(0, 4) \,.$$

We learn the optimal policy for the problem defined above using a $N = 800$-neurons wide single hidden layer neural network with ReLU nonlinearities in the mean-field regime (7) used as energy for a softmax policy (8). The initialization of the student network is as follows: first-layer weights are initialized at random drawn

independently from a centered, normal distribution with variance $\sigma^2 = 4$, while output weights are initialized at 0. The model is trained according to (4) with fixed step-size $\beta_t = 10^{-3}$. We report the results of this training procedure in Fig. 1, where we notice that all the paths monotonically decrease the error $\mathcal{E}(\nu^*) - \mathcal{E}(\nu_t)$, as predicted by our results. Note that the convergence rate of the model varies across experiments, consistently with the purely qualitative nature of the convergence result we proved.

## 6 CONCLUSIONS AND FUTURE WORK

This work addresses the problem of optimality of policy gradient algorithms, a workhorse of deep reinforcement learning, when combined with mean-field models such as neural networks. More specifically, we provide a mean-field formulation of the parametric dynamics of policy gradient algorithms for entropy-regularized MDPs and prove that, under mild assumptions, all fixed points of such dynamics are optimal. This extends similar results obtained in the "neural" or "lazy" regime to the mean-field one, which is known to be much more expressive (E et al., 2019; Ghorbani et al., 2020), but also highly nonlinear. The latter feature prevents, at present, from obtaining convergence results of these models, except in very specific settings (Chizat, 2019; Javanmard et al., 2019).

Interesting avenues or future research include the relaxing the adiabaticity assumption, *i.e.*, considering the stochastic approximation problem resulting from the finite number of samples and the finite gradient step-size, as well as establishing quantitative bounds for models with a large, but finite, number of parameters. Probably the most important open question, however, concerns establishing quantitative convergence of mean-field dynamics of neural networks: even in the supervised setting, despite recent results in specific settings (Chizat, 2019; Javanmard et al., 2019), these guarantees remain mainly out of reach.

ACKNOWLEDGMENTS.

AA acknowledges the support of the Swiss National Science Foundation through the grant P2GEP2-17501 and by the NSF grant DMS-1613337. The work of JL is in part supported by the US National Science Foundation via grants CCF-1910571 (Duke TRIPODS) and DMS-2012286.

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

## A   DERIVATION OF SOFTMAX POLICY GRADIENT DYNAMICS

**Lemma A.1.** *The gradient of the entropy-regularized value function can be written as*

$$\mathbb{E}_{s\sim\varrho_0}\left[\nabla_w V_\pi(s)\right] = \mathbb{E}_{s\sim\varrho^\pi, a\sim\pi_w}\left[\nabla_w\log\pi_w(s,a)\left(Q_\pi(s,a) - \tau\log\pi_w(s,a)\right)\right],$$

*and thus the policy gradient dynamics (5).*

*Proof.* We choose throughout $\bar{\pi}$ as the Lebesgue measure, and use that $\pi_w$ is absolutely continuous wrt $\bar{\pi}$. Taking the gradient of (1) using a parametric policy $\pi_w$ we obtain

$$
\begin{aligned}
\nabla_w V_{\pi_w}(s) &= \nabla_w \mathbb{E}_{\pi_w}\left[\sum_{t=0}^\infty \gamma^t\left(r(s_t, a_t, s_{t+1}) - \tau D_{\mathrm{KL}}(\pi_w(s_t,\cdot); \bar{\pi}(\cdot))\right)\Big| s_0 = s\right]\\
&= \nabla_w \int_{\mathcal{S}\times\mathcal{A}}\left(r(s, a_1, s_1) - \tau\log\frac{\pi_w(s, a_1)}{\bar{\pi}(a_1)} + \gamma V_{\pi_w}(s_1)\right)P(s, a_1, \mathrm{d}s_1)\pi_w(s, \mathrm{d}a_1)\\
&= \int_{\mathcal{S}\times\mathcal{A}}\left(r(s, a_1, s_1) - \tau\log\frac{\pi_w(s, a_1)}{\bar{\pi}(a_1)} + \gamma V_{\pi_w}(s_1)\right)P(s, a_1, \mathrm{d}s_1)\nabla_w\pi_w(s, \mathrm{d}a_1)\\
&\qquad + \int_{\mathcal{S}\times\mathcal{A}}\left(-\tau\nabla_w\log\frac{\pi_w(s, a_1)}{\bar{\pi}(a_1)} + \gamma\nabla_w V_{\pi_w}(s_1)\right)P(s, a_1, \mathrm{d}s_1)\pi_w(s, \mathrm{d}a_1)
\end{aligned}
\tag{A.1}
$$

Now, since $\int_{\mathcal{S}} P(s, a, \mathrm{d}s') = 1$ and $\int_{\mathcal{A}} \pi_w(s, \mathrm{d}a') = 1$ for all $s, a, w$ we have for the first term in brackets in the last line

$$\int_{\mathcal{S}\times\mathcal{A}}\nabla_w\log\frac{\pi_w(s, a_1)}{\bar{\pi}(a_1)}P(s, a_1, \mathrm{d}s_1)\pi_w(s, \mathrm{d}a_1) = \int_{\mathcal{A}}\nabla_w\pi_w(s, a_1)\mathrm{d}a_1 = \nabla_w\int_{\mathcal{A}}\pi_w(s, a_1)\mathrm{d}a_1 = 0$$

On the other hand, we can rewrite the second term in brackets as $\mathbb{E}_{\pi_w}[\gamma\nabla V_{\pi_w}(s_1)|s_0 = s]$, and recognize the LHS of (A.1) evaluated at the next state $s_1$ in the expectation. Therefore, we can sequentially repeat the same computation as above, and recalling the definition of $\varrho^{\pi_w}(\cdot)$ and $Q_\pi(s, a)$ in (6) and (2) we obtain

$$
\begin{aligned}
\nabla_w V_{\pi_w}(s) &= \int_{\mathcal{A}}\sum_{t=0}^\infty\gamma^t\left(r(s_t, a_t, s_{t+1}) - \tau\log\frac{\pi_w(s_t, a_t)}{\bar{\pi}(a_t)} + \gamma V_{\pi_w}(s_{t+1})\right)\nabla_w\pi_w(s_t, \mathrm{d}a_t)\Big|_{s_0=s}\\
&= \mathbb{E}_{\pi_w}\left[\sum_{t=0}^\infty\gamma^t\left(Q_{\pi_w}(s_t, a_t) - \tau\log\frac{\pi_w(s_t, a_t)}{\bar{\pi}(a_t)}\right)\nabla_w\log\pi_w(s_t, a_t)\Big| s_0 = s\right]\\
&= \int_{\mathcal{S}\times\mathcal{A}}\left(Q_{\pi_w}(s_t, a_t) - \tau\log\frac{\pi_w(s, a)}{\bar{\pi}(a)}\right)\nabla_w\log\pi_w(s, a)\,\varrho^{\pi_w}(\mathrm{d}s)\pi_w(s, \mathrm{d}a),
\end{aligned}
$$

where in the second line we have used that, if $\pi_w > 0$ on $\mathcal{A}$, $\pi_w(s_t, \mathrm{d}a_t)\nabla_w\log\pi_w(s_t, a_t) = \nabla_w\pi_w(s_t, \mathrm{d}a_t)$. □

**Proposition 2.1.** *For a fixed initial distribution $\varrho_0$, the dynamics (10) is the Wasserstein gradient flow of the energy functional*

$$\mathcal{E}[\nu] = \mathbb{E}_{s_0\sim\varrho_0}\left[V_{\pi_\nu}(s_0)\right].$$

*Proof.* We find the potential of the gradient flow by functional differentiation of $\mathcal{E}$:

$$\frac{\delta}{\delta\nu}\mathcal{E}[\nu](\omega) = \int_{\mathcal{A}}\frac{\delta\mathcal{E}}{\delta\pi}(s, a)\frac{\delta\pi_\nu}{\delta\nu}(s, a; \omega)\mathrm{d}s\,\mathrm{d}a \tag{A.2}$$

and consider the two terms in the integral separately, starting from the second:

$$
\begin{aligned}
\frac{\delta\pi_\nu}{\delta\nu}(s, a; \omega) &= \frac{\delta}{\delta\nu}\frac{e^{\tau\int\psi(s,a;\omega)\nu(\mathrm{d}\omega)}}{\int_{\mathcal{A}}e^{\tau\int\psi(s,a;\omega)\nu(\mathrm{d}\omega)}\mathrm{d}a}\\
&= \frac{1}{\int_{\mathcal{A}}e^{\tau\int\psi(s,a;\omega)\nu(\mathrm{d}\omega)}\mathrm{d}a}\frac{\delta}{\delta\nu}e^{\tau\int\psi(s,a;\omega)\nu(\mathrm{d}\omega)}
\end{aligned}
$$

$$-\frac{e^{\tau\int\psi(s,a;\omega)\nu(\mathrm{d}\omega)}}{\left(\int_{\mathcal{A}}e^{\tau\int\psi(s,a;\omega)\nu(\mathrm{d}\omega)}\mathrm{d}a\right)^{2}}\int_{\mathcal{A}}\frac{\delta}{\delta\nu}e^{\tau\int\psi(s,a;\omega)\nu(\mathrm{d}\omega)}\mathrm{d}a$$

$$=\tau\left(\psi(s,a;\omega)-\int_{\mathcal{A}}\psi(s,a';\omega)\pi_{\nu}(s,\mathrm{d}a')\right)\pi_{\nu}(s,a)\tag{A.3}$$

For the first term in the integrand of (A.2), we use $\pi(s,a)$ as a density and obtain

$$\frac{\delta\mathcal{E}}{\delta\pi}(s,a)=\frac{\delta}{\delta\pi}\left[\iint_{\mathcal{S}\times\mathcal{A}}\left(\bar{r}(s,a)-\tau\log\pi(s,a)\right)\left(\varrho_{\pi}(\mathrm{d}s)\pi(s,\mathrm{d}a)\right)\right](s,a)$$

$$=\int_{\mathcal{S}}\left[\left(\bar{r}(s,a)-\tau(\log\pi(s,a)+1)\right)\varrho^{\pi}(s',s)\right.\tag{A.4}$$

$$\left.+\int_{\mathcal{S}\times\mathcal{A}}\left(\bar{r}(s'',a'')-\tau\log\pi(s'',a'')\right)\frac{\delta\varrho^{\pi}(s',s'')}{\delta\pi}(s,a)\pi(s'',\mathrm{d}a'')\mathrm{d}s''\right]\varrho_{0}(\mathrm{d}s')\,,\tag{A.5}$$

where in the last line we have used that $\frac{\delta}{\delta\pi}[\pi\log\pi](s,a)=\log\pi(s,a)+1$.
We evaluate the variational derivative of $\varrho^{\pi}$ as

$$\frac{\delta\varrho^{\pi}(s',s'')}{\delta\pi}(s,a)=\sum_{t=0}^{\infty}\gamma^{t}\frac{\delta}{\delta\pi}\left[\int P_{\pi}^{t}(s',s'')\right](s,a)$$

$$=\gamma\int_{\mathcal{S}}\frac{\delta P_{\pi}(s',s''')}{\delta\pi}\left(\sum_{t=0}^{\infty}\gamma^{t}P_{\pi}^{t}(s''',s'')\right)+P_{\pi}(s',s''')\frac{\delta}{\delta\pi}\sum_{t=0}^{\infty}\gamma^{t}\frac{\delta}{\delta\pi}P_{\pi}^{t}(s''',s'')\mathrm{d}s'''$$

$$=\gamma\left[\int_{\mathcal{S}}\delta(s,s')P(s',a,s''')\varrho^{\pi}(s''',s'')+P_{\pi}(s',s''')\frac{\delta\varrho^{\pi}(s''',s'')}{\delta\pi}(s,a)\mathrm{d}s'''\right]$$

We further recognize the same derivative on the RHS of the above expression, allowing to write

$$\frac{\delta\varrho^{\pi}(s',s'')}{\delta\pi}(s,a)=\sum_{t=0}^{\infty}\gamma^{t}P_{\pi}^{t}(s',s)\int_{\mathcal{S}}P(s,a,s''')\varrho^{\pi}(s''',s'')\mathrm{d}s'''$$

Finally, we notice that the last term in (A.4) is constant in $a$ and therefore vanishes when integrated against (A.3), so we only consider

$$\frac{\delta\mathcal{E}}{\delta\pi}(s,a)-\varrho_{\pi}(s)=\int_{\mathcal{S}^{2}\times\mathcal{A}}\left(\bar{r}(s'',a'')-\tau\log\pi(s'',a'')\right)\times$$

$$\times\left(\delta_{s,s''}\delta_{a,a''}\varrho^{\pi}(s',s)+\pi(s'',a'')\frac{\delta\varrho^{\pi}(s',s'')}{\delta\pi}(s,a)\right)\varrho_{0}(\mathrm{d}s')\mathrm{d}s''\mathrm{d}a''$$

$$=\int_{\mathcal{S}^{2}\times\mathcal{A}}\sum_{t=0}^{\infty}\gamma^{t}P_{\pi}^{t}(s',s)\left(\bar{r}(s'',a'')-\tau\log\pi(s'',a'')\right)\times$$

$$\times\left(\delta_{s,s''}\delta_{a,a''}+\gamma\int_{\mathcal{S}}P(s,a,s''')\varrho^{\pi}(s''',s'')\mathrm{d}s'''\right)\varrho_{0}(\mathrm{d}s')\mathrm{d}s''\mathrm{d}a''$$

$$=\sum_{t=0}^{\infty}\gamma^{t}\int_{\mathcal{S}}P_{\pi}^{t}(s',s)\left(Q_{\pi}(s,a)-\tau\log\pi(s,a)\right)\varrho_{0}(\mathrm{d}s')$$

$$=\left(Q_{\pi}(s,a)-\tau\log\pi(s,a)\right)\varrho_{\pi}(s)\tag{A.6}$$

We conclude by noting that combining (A.3) and (A.6) we obtain $C_{\pi}[\psi(\omega),Q_{\pi}-\tau\log\pi](s)$, and the Wasserstein gradient flow corresponding to this potential is (10). □

## B  PROOFS OF THE MANY-PARTICLE LIMIT

**Theorem 4.1.** *Let Assumption 1 hold and let $w_{t}^{(N)}$ be a solution of (5) with initial condition $w_{0}^{(N)}\in\mathcal{W}=\Omega^{N}$. If $\nu_{0}^{(N)}$ converges to $\nu_{0}\in\mathcal{P}_{2}(\Omega)$ in Wasserstein distance $W_{2}$ then $\nu_{t}^{(N)}$ converges, for every $t>0$, to the unique solution $\nu_{t}$ of (10).*

*Proof.* As anticipated in the main text, the proof is divided in two parts:

1. We prove sufficient regularity of the dynamics (10), allowing to establish existence and uniqueness of its solution.

2. We leverage the regularity proven above to establish a propagation of chaos result, showing that the system of interacting particles behaves asymptotically as its mean-field limit.

While carrying out this proof is needed in our context since the dependence of $\mathcal{E}(\nu)$ on $\nu$ is more involved than in *e.g.*, Mei et al. (2018); Rotskoff & Vanden-Eijnden (2018); Chizat & Bach (2018), the steps of this derivation are mainly standard, see e.g., (Sznitman, 1991).

### B.1 REGULARITY

We prove existence and uniqueness of the gradient flow dynamics (10) through standard arguments from the optimal transportation literature (see *e.g.*, (Ambrosio et al., 2008)). More specifically, recalling that $\pi = \pi_\nu$ we leverage the Lipschitz continuity of the vector field

$$F_t(\omega, \nu) = \int_{\mathcal{S}} C_\pi \left[ \nabla_\omega \psi(s, \cdot; \omega), Q_\pi(s, \cdot) - \tau \log \pi(s, \cdot) \right] \varrho_\pi(\mathrm{d}s)$$

with respect to $\nu$. To prove such regularity result, decompose

$$\mathcal{E}(\nu) = R\left( \int \psi \nu \right) \qquad \text{for} \quad R(f) = S \circ \pi(f). \tag{B.1}$$

and $S : (\mathcal{S} \to \mathcal{M}_+^1(\mathcal{A})) \to \mathbb{R}$ maps $\mu \mapsto \mathbb{E}_\mu \left[ V_\mu(s) | s \sim \varrho_0 \right]$ and $\pi : L^2(\mathcal{S} \times \mathcal{A}) \to (\mathcal{S} \to \mathcal{M}_+^1(\mathcal{A}))$ is the softmax policy parametrization (8) of its argument. Recalling the definition of $\mathcal{Q}_r$ from Assumption 1 and denoting $\mathcal{F}_r = \{ \int \psi \nu : \operatorname{supp} \nu \in \mathcal{Q}_r \}$, we further define the norms and constants needed in the following proof as:

$$\|D\psi\|_{r,\infty} = \sup_{\omega \in \mathcal{Q}_r} \|D\psi_\omega\| \qquad\qquad L_{D\psi} = \sup_{\omega, \omega' \in \mathcal{Q}_r} \frac{\|D\psi_\omega - D\psi_{\omega'}\|}{\|\omega - \omega'\|_2}$$

$$\|DR\|_{r,\infty} = \sup_{\psi(\cdot; \omega) : \omega \in \mathcal{Q}_r} \|DR_\psi\| \qquad L_{DR} = \sup_{\psi, \psi' \in \mathcal{F}_r} \frac{\|DR_\psi - DR_{\psi'}\|}{\|\psi - \psi'\|_2}$$

where $\| \cdot \|$ denotes the operator norm. While for any $r > 0$ the boundedness of $\|D\psi\|_{r,\infty}$, $L_{D\psi}$ results directly from Assumption 1, more work is needed to prove that $\|DR\|_{r,\infty} < \infty$, $L_{DR} < \infty$. We prove this in Lemma C.5 below, and proceed with the proof of convergence of the particle dynamics.

For any $r$ and corresponding $\mathcal{Q}_r$ from Assumption 1 we define the set of localized functionals

$$\mathcal{E}^{(r)}(\nu) = \begin{cases} \mathcal{E}(\nu) & \text{if } \operatorname{supp}(\nu) \subset \mathcal{Q}_r \\ \infty & \text{else} \end{cases}$$

Furthermore, we say that a coupling $\gamma \in \mathcal{M}_+^1(\Omega \times \Omega)$ is an *admissible transport plan* if both its marginals have support in $\mathcal{Q}_r$ and finite second moments. To every admissible transport plan, for $p \geq 1$ we associate a *transportation cost* $C_p(\gamma) = (\int_{\Omega^2} |\omega - \omega'|^p \mathrm{d}\gamma(\omega, \omega'))^{1/p}$.

We prove the following results: for every $r > 0$ we have

1. There exists $\lambda_r > 0$ such that for any admissible transport plan $\gamma$, defining the interpolation map $\nu_t^\gamma := (t\Pi_0 + (1-t)\Pi_1)_\# \gamma$, the function $t \mapsto \mathcal{E}(\nu_t^\gamma)$ is differentiable with Lipschitz continuous derivative with constant $\lambda_r C_2^2(\gamma)$.

2. Let $\nu_0$ have support in $\mathcal{Q}_r$. Then for any given transport plan $\gamma$ with first marginal given by $\nu_0$, a velocity field $F$ satisfies

$$\mathcal{E}((\Pi_1)_\# \gamma) \geq \mathcal{E}(\nu_0) + \int F(u) \cdot (u - u') \mathrm{d}\gamma(u, u') + o(C_2(\gamma)) \tag{B.2}$$

if and only if $F(u)$ is in the subdifferential of $D\mathcal{E}_\nu(u) := C_\pi[\psi, Q_\pi - \log \pi](u)$ for $g \in L^2(\mathcal{S} \times \mathcal{A})$ (projected in the interior of $\mathcal{Q}_r$ when $u \in \partial \mathcal{Q}_r$ ) for $\nu_0$ almost every $u \in \Omega$.

The proof of the two points above corresponds to (Chizat & Bach, 2018, Lemma B.2). We sketch the proof of these two points below, referring to the original reference for the details.

1. By the Lipschitz continuity of $\psi : \Omega \to L^2(\mathcal{S} \times \mathcal{A})$ and $R'(f) = DR_f = C_\pi[\cdot, Q_\pi - \tau \log \pi]$ in $\mathcal{Q}_r$, the energy $\mathcal{E}^{(r)}(\nu_t^\gamma)$ transported along an interpolating path $\nu_t^\gamma$ is differentiable and we can write its derivative as

$$\frac{\mathrm{d}}{\mathrm{d}t}\mathcal{E}^{(r)}(\nu_t^\gamma) = \int R'(\int \psi \nu_t^\gamma) \int D\psi_{(1-t)\omega+t\omega'}(\omega' - \omega)\mathrm{d}\gamma(\omega, \omega')$$

Then, again by the Lipschitz continuity of $D\psi$ and $DR$ we have for $0 \le t' < t'' < 1$

$$\left| \frac{\mathrm{d}}{\mathrm{d}t}\mathcal{E}^{(r)}(\nu_{t'}^\gamma) - \frac{\mathrm{d}}{\mathrm{d}t}\mathcal{E}^{(r)}(\nu_{t''}^\gamma) \right| \le \left| \int (R'(\int \psi \nu_{t'}^\gamma) - R'(\int \psi \nu_{t''}^\gamma)) \int D\psi_{(1-t')\omega+t'\omega'}(\omega' - \omega)\mathrm{d}\gamma(\omega, \omega') \right|$$

$$+ \left| \int R'(\int \psi \nu_{t''}^\gamma) \int (D\psi_{(1-t')\omega+t'\omega'} - D\psi_{(1-t'')\omega+t''\omega'})(\omega' - \omega)\mathrm{d}\gamma(\omega, \omega') \right|$$

$$\le \lambda_r |t'' - t'|$$

for a $\lambda_r$ large enough, where in the last inequality we have used the uniform bounds on $DR$ in $\mathcal{Q}_r$, that $|D\psi_{(1-t')\omega+t'\omega'} - D\psi_{(1-t'')\omega+t''\omega'}| \le (t' - t'')L_{D\psi}|\omega - \omega'|$ and we applied Hölder's inequality to bound $C_1^2(\gamma) \le C_2^2(\gamma)$.

2. The proof of this result leverages an expansion of the functionals $R, \psi$ to the second order in their arguments:

$$\psi(\omega') = \psi(\omega) + D\psi_\omega(\omega' - \omega) + \mathcal{R}_\psi(\omega, \omega')$$
$$R(g) = R(f) + DR_f(g - f) + \mathcal{R}_R(f, g)$$

Recalling the Lipschitz bounds on the remainders $\mathcal{R}_\psi(\omega, \omega') < \frac{1}{2}L_{D\psi}|\omega - \omega'|^2$, $\mathcal{R}_R(f, g) < \frac{1}{2}L_{DR}\|f - g\|_2^2$ and combining the two expansions above we have, for a transport plan $\gamma$ with marginals $\nu_0 = (\Pi_0)_\# \gamma$, $\nu_1 = (\Pi_1)_\# \gamma$

$$\mathcal{E}^{(r)}(\nu_1) = \mathcal{E}^{(r)}(\nu_0) + \int R'(\int \psi \nu_0) D\psi_u(u' - u)\mathrm{d}\gamma(u, u')\mathrm{d}s\mathrm{d}a + \mathcal{R}$$

for a remainder term $\mathcal{R}$. We can bound such remainder term, again by the Lipschitz regularity of $D\psi$ and $DR$ and by the boundedness of $D\psi, DR$ in $\mathcal{Q}_r$, by $C_2(\gamma)^2$ and $C_1(\gamma)^2 \le C_2(\gamma)^2$, thereby obtaining that

$$\mathcal{E}^{(r)}(\nu_1) = \mathcal{E}^{(r)}(\nu_0) + \int R'(\int \psi \nu_0) D\psi_u(u' - u)\,\mathrm{d}s\mathrm{d}a\,\mathrm{d}\gamma(u, u') + o(C_2(\gamma))$$

Noting that the integrand against the coupling is the gradient flow vector field, the above uniquely characterizes the velocity field satisfying (B.2).

We note that point 1) above immediately implies that $\mathcal{E}(\cdot)$ is $\lambda_r$-semiconvex along geodesics, while by 2) $\mathcal{E}^{(r)}(\cdot)$ admits strong Wasserstein subdifferentials on its domain (Ambrosio et al., 2008, Definition 10.3.1). Combining the two results one obtains existence and uniqueness of the solutions of the Wasserstein gradient flow through (Ambrosio et al., 2008, Theorem 11.2.1).

### B.2 PROPAGATION OF CHAOS

By the Lipschitz continuity of the transport field in (10) in $\nu_0$ with $\mathrm{supp}\, \nu_0 \subset \mathcal{Q}_r$, there exists a time $t_r > 0$ such that, $\mathrm{supp}\, \nu_s^{(N)} \subset \mathcal{Q}_r$ for all $s \in [0, t_r]$, $N \in \mathbb{N}$. Consider now two times $0 \le t_1 < t_2 \le t_r$. To prove the existence of the limiting curve $(\nu_t)_t$, we show that the curves $\nu_t^{(N)}$ are uniformly in $N$ equicontinuous in $W_2$ and as such, possess a converging subsequence by Arzela-Ascoli theorem. To show equicontinuity, we bound the the $W_2$ distance between distributions by coupling positions of the same particles at different times and using Cauchy-Schwartz inequality:

$$W_2(\nu_{t_1}^{(N)}, \nu_{t_2}^{(N)})^2 \le \frac{1}{N} \sum_{i=1}^N \|w_{t_1}^{(i)} - w_{t_2}^{(i)}\|_2^2 \le \frac{t_2 - t_1}{N} \sum_{i=1}^N \int_{t_1}^{t_2} \|\frac{\mathrm{d}}{\mathrm{d}s} w_s^{(i)}\|_2^2 \mathrm{d}s$$

Combining the above with the identity

$$\frac{\mathrm{d}}{\mathrm{d}t}\mathcal{E}(\nu_t^{(N)}) = \frac{1}{N}\sum_{i=1}^{N}\langle\nabla_{w_i}\mathcal{E}(\nu_t^{(N)}), \frac{\mathrm{d}}{\mathrm{d}t}w_t^{(i)}\rangle = \frac{1}{N}\sum_{i=1}^{N}\|\frac{\mathrm{d}}{\mathrm{d}t}w_t^{(i)}\|_2^2$$

we have

$$W_2(\nu_{t_1}^{(N)}, \nu_{t_2}^{(N)}) \leq \sqrt{t_2 - t_1}\sqrt{\int_{t_1}^{t_2}\frac{\mathrm{d}}{\mathrm{d}s}\mathcal{E}(\nu_s^{(N)})\mathrm{d}s} \leq \sqrt{t_2 - t_1}\left(\sup_{\mathrm{supp}\,\nu\in Q_r}\mathcal{E}(\nu) - \inf_{\mathrm{supp}\,\nu\in\mathcal{Q}_r}\mathcal{E}(\nu)\right)^{1/2}$$

where we recall that $\mathcal{F}_r = \{\nu \in \mathcal{M}_+^1(\Omega) : \mathrm{supp}\,\nu \subset \mathcal{Q}_r\}$. In particular, the above continuity bound is independent on $N$, proving equicontinuity of $\nu^{(N)}$ in $W_2$.

We now prove that the limiting point of the converging subsequence whose existence was identified above must solve (10). To do so we compare both the differential for the mean-field and particle dynamics to the one of a linear inhomogeneous PDE: for any bounded and continuous $f : \mathbb{R} \times \mathbb{R}^m \to \mathbb{R}^m$, denoting by $E = \nu_t F_t\mathrm{d}t$ and $E_N = \nu_t^{(N)}F_t^{(N)}\mathrm{d}t$ where $F_t, F_t^{(N)}$ are the vector fields of the mean-field and particle system respectively, we write

$$\left|\int f(\omega)\mathrm{d}(E - E_N)\right|_2 \leq \|f\|_\infty\int\left|F_t^{(N)} - F_t\right|_2\mathrm{d}\nu_t^{(N)}\mathrm{d}t + \left|\int fF_t\mathrm{d}(\nu_t^{(N)} - \nu_t)\mathrm{d}t\right|.$$

By boundedness of $fF_t$ over $\mathcal{Q}_r$, the second term converges by our choice of subsequence. For the first term, denoting throughout by $\|\cdot\|_{BL}$ the bounded Lipschitz norm, we leverage the Lipschitz continuity of the vector field $F$ WRT the underlying parametric measure:

$$\|F_t^{(N)} - F_t\|_2 \leq C_r\|\nu_t - \nu_t^{(N)}\|_{BL}$$

for $C_r > 0$ large enough, again obtaining convergence by our choice of subsequence. This proves convergence of the particle model to the mean-field equation (10) on $[0, t_r]$. To extend the time interval on which we prove convergence, we use that $\mathcal{E}(\nu)$ (and thus $R$) decays along trajectories of (10). Consequently, by the boundedness of the differential $DR$ on sublevel sets of $R$ the Lipschitz constant of $\mathcal{E}(\nu_t)$ is uniformly bounded. Using again the Lipschitz continuity of $F$ we can show that $\sup_{u\in\mathcal{Q}_r}\|F\| < A + Br$, *i.e.*, that particle velocities can grow at most linearly in $r$, and an application of Gronwall allows to find, for every $T > 0$ that there exists $r > 0$ such that $\mathrm{supp}\,\nu_t \in \mathcal{Q}_r$ for all $t \in [0, T]$ and propagation of chaos follows. $\qquad\square$

## C  PROOFS OF OPTIMALITY

**Theorem 4.2.** *Let Assumption 1 hold and $\nu_t$ given by* (10) *converge to $\nu^*$, then $\pi_{\nu^*} = \pi^*$ $\mathcal{S}\times\mathcal{A}$-a.e.*

Before proceeding to prove Theorem 4.2, we state the alternative form of Assumption 1 a) in the case where $\Theta \neq \mathbb{R}^{m-1}$. Our proof of the theorem above can be easily generalized to the setting of Assumption 2.

**Assumption 2.** *Assume that $\omega = (\omega_0, \bar{\omega}) \in \mathbb{R} \times \Theta$ for $\Theta \subset \mathbb{R}^{m-1}$ which is the closure of a bounded open convex set. Furthermore $\psi(s, a; \omega) = \omega_0\phi(s, a; \bar{\omega})$ where $\phi$ is bounded, differentiable and $D\phi_\omega$ is Lipschitz. Also, for all $f \in L^2(\mathcal{S}\times\mathcal{A})$ the regular values of the map $\bar{\omega} \mapsto g_f(\bar{\omega}) := \int f(s, a)\phi(s, a; \bar{\omega})\mathrm{d}a\mathrm{d}s$ are dense in its range and $g_f(\bar{\omega})$ satisfies Neumann boundary conditions (i.e., for all $\bar{\omega} \in \partial\Theta$ we have $dg_f(\bar{\omega})(n_{\bar{\omega}}) = 0$ where $n_{\bar{\omega}} \in \mathbb{R}^{m-1}$ is the normal of $\partial\Theta$ at $\bar{\omega}$).*

We prove Theorem 4.2 as sketched in Section 4 by first connecting the optimality and the support of stationary measures in parametric space through Lemma C.1, and then investigating how the dynamics preserves full support property for any $t > 0$ in Lemma C.2 and avoids spurious minima in Lemma C.3.

Before starting this program we introduce the equivalent of greedy policies in the entropy-regularized setting. For a given $Q(s, a)$, the associated *Boltzmann policy* $\pi_\mathcal{B}$ with respect to a reference measure $\bar{\pi}$ is given by

$$\pi_\mathcal{B}(s, a) := \exp\left[(Q(s, a) - V_Q(s))/\tau\right] \quad \text{for} \quad V_Q(s) := \tau\log\mathbb{E}_{a\sim\bar{\pi}}\left[\exp\left[Q(s, a)/\tau\right]\right],$$

and satisfies

$$\pi_\mathcal{B}(s, \cdot) = \arg\max_{\pi\in\mathcal{M}_+^1(\mathcal{A})}\left(\mathbb{E}_{a\sim\pi}\left[Q(s, a)\right] - \tau D_{\mathrm{KL}}(\pi; \bar{\pi})(s)\right)$$

One can then define the *Boltzmann backup* or *soft Bellman backup* operator $T^\tau$ that, for a given $Q$ and the associated Boltzmann policy $\pi_\mathcal{B}$, gives the action-value function $T^\tau Q$ associated to $\pi_\mathcal{B}$:

$$T^\tau Q(s, a) = \bar{r}(s, a) + \gamma\tau\mathbb{E}_\pi\left[\log\mathbb{E}_{a_1\sim\bar{\pi}}\left[\exp Q(s_1, a_1)/\tau\right]|s_0 = s\right] \qquad (C.1)$$

It is known (Haarnoja et al., 2018b, Theorem 1) that the fixed points of the above operator are optimal, *i.e.*, they correspond to the optimal policy $\pi_{\mathcal{B}}^* = \pi^*$ of the entropy-regularized MDP.

To state the first partial result towards the proof of Theorem 4.2, we observe that the $\omega_0$-component of the transport vector field in (10) can be written as

$$\left( \int_{\mathcal{S}} C_{\pi_\nu} \left[ \nabla_\omega \psi(s, \cdot; \omega), Q_{\pi_\nu}(s, \cdot) - \tau \log \pi_\nu \right] \varrho_{\pi_\nu}(\mathrm{d}s) \right)_0 = \int_{\mathcal{S}} C_{\pi_\nu} \left[ \phi(s, \cdot; \bar{\omega}), Q_{\pi_\nu}(s, \cdot) - \tau \log \pi_\nu \right] \varrho_{\pi_\nu}(\mathrm{d}s) , \tag{C.2}$$

where we recall that $C_\pi[f, g]$ is the covariance operator WRT the probability measure $\pi(s, \mathrm{d}a)$ introduced below (10). We note in particular that the above expression only depends on $\bar{\omega}$.

With the above information at hand we now proceed to prove Lemma C.1 relating the value of (C.2) and the optimality of fixed points of (10):

**Lemma C.1.** *Let Assumption 1 hold and let $\nu$ satisfy*

$$\int_{\mathcal{S} \times \mathcal{A}} \phi(\bar{\omega}; s, a) \left( Q_{\pi_\nu}(s, a) - \tau \log \pi_\nu(s, a) - V_{\pi_\nu}(s) \right) \pi_\nu(s, \mathrm{d}a) \varrho_{\pi_\nu}(\mathrm{d}s) = 0 , \tag{C.3}$$

$\bar{\omega}$-*almost everywhere in* $\Theta$. *Then we have that* $Q_{\pi_\nu} = Q_{\pi^*}$ *holds* $\bar{\pi} \varrho_{\bar{\pi}}$-*a.e. in* $\mathcal{S} \times \mathcal{A}$.

*Proof of Lemma C.1.* Assuming that (C.3) holds Lebesgue-a.e. in $\Theta$, by the assumed continuity of $\phi$ in $\bar{\omega}$ combined with the expressivity of $\phi$ Assumption 1 b) we must have that

$$Q_{\pi_\nu}(s, a) - \tau \log \pi_\nu(s, a) - V_{\pi_\nu}(s) = 0 \qquad \pi_\nu \varrho_{\pi_\nu} - \text{a.e.} .$$

We then rewrite the above condition in compact notation as the fixed point equation

$$T^\tau Q_{\pi_\nu}(s, a) = Q_{\pi_\nu}(s, a)$$

for the *soft Q learning* or *Boltzmann backup* operator $T^\tau$ defined in (C.1). Since all fixed points of $T^\tau$ for $\gamma < 1$ are optimal (Nachum et al., 2017, Theorem 3), we must have that $\pi_\nu = \pi_{\mathcal{B}}[Q^*] = \pi^*$ for $\pi_\nu \varrho_{\pi_\nu}$-almost every $(s, a) \in \mathcal{S} \times \mathcal{A}$. The result follows by equivalence of $\pi_\nu$ and $\bar{\pi}$. $\square$

Consequently, suboptimal fixed points of the dynamics (10) cannot satisfy (C.3) Lebesgue-a.e. in $\Theta$.

## C.1 PROOF OF THEOREM 4.2

We prove below that spurious local minima that do not satisfy (C.3) $\Theta$-a.e. are avoided by the dynamics. We do so by leveraging the approximate gradient structure of the policy gradient vector field when $\nu_t$ is close to one of such stationary points, as discussed in the main text. Combining this fact with the assumed convergence to $\nu^*$ proves Theorem 4.2.

We note that by the assumed homogeneity of $\psi$ in its first component, if $\nu, \nu'$ are such that

$$\int \omega_0 \nu(\mathrm{d}\omega_0, \mathrm{d}\bar{\omega}) = \int \omega_0 \nu'(\mathrm{d}\omega_0, \mathrm{d}\bar{\omega}) \quad \text{a.e.} \tag{C.4}$$

then

$$f_\nu(\cdot) = \int \omega_0 \phi(\cdot; \bar{\omega}) \nu(\mathrm{d}\omega_0, \mathrm{d}\bar{\omega}) = \int \omega_0 \phi(\cdot; \bar{\omega}) \nu'(\mathrm{d}\omega_0, \mathrm{d}\bar{\omega}) = f_{\nu'}(\cdot) ,$$

so that in turn we have $\pi_\nu = \pi_{\nu'}$ a.e.. In other words, the homogeneity of the chosen class of approximators results in a degeneracy of the map $\psi : \mathcal{M}_+^1(\Omega) \mapsto L^2(\mathcal{S} \times \mathcal{A})$. To remove this degeneracy in our analysis, we identify all the distributions $\nu, \nu'$ that are equivalent under (C.4) by defining the signed measure

$$h_\nu^1(\mathrm{d}\bar{\omega}) := \int \omega_0 \nu(\mathrm{d}\omega_0, \mathrm{d}\bar{\omega}) \tag{C.5}$$

Leveraging this definition, we prove the desired result Theorem 4.2 in two key steps: we show that

1. the solution to (10) does not lose (projected) support for any finite time, thereby preserving the property from Assumption 1 c),

2. stationary points $\tilde{\nu}$ with $Q_{\pi_{\tilde{\nu}}} \neq Q_{\pi^*}$ – which by Lemma C.1 cannot have full projected support in $\Theta$ – are avoided by the dynamics.

These facts are respectively summarized in the following lemmas:

**Lemma C.2.** *Let Assumption 1 a) hold and let $\nu_0$ satisfy Assumption 1 c), then for every $t > 0$, $\nu_t$ solving (10) with initial condition $\nu_0$ also satisfies Assumption 1 c).*

Throughout, we let $\| \cdot \|_{BL}$ denote the bounded Lipschitz norm.

**Lemma C.3.** *Let Assumption 1 hold and let $\tilde{\nu}$ be a fixed point of (10) such that (C.3) does not hold a.e.. Then there exists $\varepsilon > 0$ such that if $\|h_{\tilde{\nu}}^1 - h_{\nu_{t_1}}^1\|_{BL} < \varepsilon$ for a $t_1 > 0$ there exists $t_2 > t_1$ such that $\|h_{\tilde{\nu}}^1 - h_{\nu_{t_2}}^1\|_{BL} > \varepsilon$.*

*Proof of Lemma C.2.* Analogously to (Chizat & Bach, 2018, Lemma C.13), we aim to show that the separation property Assumption 1 c) is preserved by the evolution of $\nu_0$ along the characteristic curves $X(t, u)$ solving

$$\partial_t X(t, u) = F_t(X(t, u); \nu_t), \tag{C.6}$$

where $F_t$ is the transport field in (10). To reach this conclusion, the analogous result in Chizat & Bach (2018) only relies on the *continuity* of the map $u \mapsto X(t, u)$, established in (Chizat & Bach, 2018, Lemma B.4) under Assumption 1 a). Hence, it is sufficient for our purposes to establish continuity of the map $X(t, \cdot)$ from (C.6). This property, however results immediately from the one-sided Lipschitz continuity of the vector field $F_t$ on $\mathcal{Q}_r = [-r, r] \times \Theta$ uniformly on compact time intervals, which is in turn guaranteed by the Lipschitz continuity and Lipschitz smoothness of $\psi$ from Assumption 1 and boundedness of $r$. □

To simplify the notation in the following proof, we denote throughout

$$\delta(\nu) := Q_{\pi_\nu} - \tau \log \pi_\nu - V_{\pi_\nu} \qquad \text{and} \qquad \langle f, g \rangle_\pi := \int_{\mathcal{S} \times \mathcal{A}} f(s, a) g(s, a) \pi(s, \mathrm{d}a) \varrho_\pi(\mathrm{d}s).$$

*Proof of Lemma C.3.* We first claim that by Lemma C.1, for any spurious fixed point (such that $Q_{\pi_{\tilde{\nu}}} \neq Q_{\pi^*}$), there must exist a subset of $\Theta$ with positive Lebesgue measure where $\tilde{\nu}$ loses support and such that $\langle \nabla \psi, \delta(\tilde{\nu}) \rangle_\pi \neq 0$. This is easily proven by contradiction: if $\langle \nabla \psi, \delta(\tilde{\nu}) \rangle_\pi = 0$ a.e. then by Lemma C.1 we have that $Q_{\pi_{\tilde{\nu}}} = Q_{\pi^*}$. This implies that the quantity

$$g_{\tilde{\nu}}(\bar{\omega}) := \langle \partial_{\omega_0} \psi(\cdot; \omega), \delta(\tilde{\nu}) \rangle_{\pi_\nu} = \langle \psi(\cdot; (1, \bar{\omega})), \delta(\tilde{\nu}) \rangle_{\pi_\nu} = \langle \phi(\cdot; \bar{\omega}), \delta(\tilde{\nu}) \rangle_{\pi_\nu} \tag{C.7}$$

cannot vanish a.e. on $\Theta$. Then, by Assumption 1 on the regularity of $g$, there exists a nonzero regular value $-\eta$ of $g_{\tilde{\nu}}(\bar{\omega})$. Assuming without loss of generality that this regular value is negative, so that $\eta > 0$ (else invert the signs of $\omega_0$ in the remainder of the proof), we define the nonempty sublevel set $\mathcal{G} := \{(\omega_0, \bar{\omega}) \in \Omega : g_{\tilde{\nu}}(\bar{\omega}) < -\eta\}$ and

$$\mathcal{G}_+ = \{(\omega_0, \bar{\omega}) \in \mathcal{G} : \omega_0 > 0\}. \tag{C.8}$$

Further denoting by $\bar{G} \subseteq \Theta$ the projection of $\mathcal{G}$ onto $\Theta$, we have by definition that the gradient field of $g_{\tilde{\nu}}(\bar{\omega})$ is orthogonal to the level set $\partial \bar{G}$, the latter being an orientable manifold of dimension $m - 2$. Denoting by $n_{\bar{\omega}}$ the normal unit vector to $\partial \bar{G}$ in the outward direction, by continuity of $\nabla g_{\tilde{\nu}}(\bar{\omega})$ when $\bar{G}$ is compact[1] we can bound the scalar product between the two away from 0, *i.e.*, there exists

$$\beta := \min_{\bar{\omega} \in \partial \bar{G}} n_{\bar{\omega}} \cdot \nabla_{\bar{\omega}} g_{\tilde{\nu}}(\bar{\omega}) > 0.$$

We now prove that the stationarity assumption in a $\varepsilon$-neighborhood of a spurious fixed point

$$\|h_{\tilde{\nu}}^1 - h_{\nu_t}^1\|_{BL} < \varepsilon \qquad \text{for all } t > t_1, \tag{C.9}$$

leads to a contradiction for $\varepsilon$ small enough. To do so, by Lemma C.5 we set $\varepsilon(\alpha, \eta, \beta)$ small enough so that for all $\nu_t$ such that (C.9) holds we have $g_{\nu_t}(\bar{\omega}) < -\eta/2$ on $\bar{G}$ and $n_{\bar{\omega}} \cdot \nabla_{\bar{\omega}} g_{\nu_t} > \beta/2$ on $\partial \bar{G}$. Then, the two inequalities above combined with $\partial_{\omega_0} \psi(\omega_0, \bar{\omega}) = \psi(1, \bar{\omega})$ imply that the set $\mathcal{G}_+$ defined above is forward invariant and therefore that $\partial_t \nu_t(\mathcal{G}_+) \geq 0$ as long as (C.9) holds. Furthermore, by similar arguments we notice that characteristic trajectories cannot enter the set $\mathcal{G} \setminus \mathcal{G}_+$ after $t_1$.

Now, we consider two cases: either (i) a positive amount of mass is present at $t_1$ in the forward invariant set $\mathcal{G}_+$ ($\nu_{t_1}(\mathcal{G}_+) > 0$) or (ii) $\nu_{t_1}(\mathcal{G}_+) = 0$. We discuss these two cases separately, along the lines of (Chizat & Bach, 2018, Lemmas C.4, C.18), respectively.

---

[1] if $\bar{G}$ is not compact we choose $\eta$ to also be a regular value of the function on $\{\bar{\omega} \in \mathbb{R}^{m-1} : \|\bar{\omega}\|_2 = 1\}$ to which $g$ converges as $\bar{\omega}$ goes to infinity.

(i) Assume that $\nu_{t_1}(\mathcal{G}_+) > 0$. We note that under our assumptions the first component of the velocity field in $\mathcal{G}$ is lower bounded by $\eta/2$, so that $\omega_0(t) = \omega_0(0) + t\eta/2$ bounds from below the $\omega_0$-component of the trajectory of a test mass with initial condition with $\omega(0) \in \mathcal{G}$, as long as $\bar{\omega}(t) \in \bar{G}$. Combining this bound with the forward invariance of $\mathcal{G}_+$, we see that if $\omega(0) \in \mathcal{G}_+$ then $\omega_0(t) > t\eta/2$. Consequently, assuming that supp$(\nu_t) \subset (-M, M) \times \Theta$ for every $t > t_1$ we have

$$h_{\nu_t}^1(\bar{G}) \geq \eta/2(t - t_1)\nu_{t_1}(\mathcal{G}_+) + \min\{0, (t - t_1)\eta/2 - M\}\nu_{t_1}(\mathcal{G} \setminus \mathcal{G}_+).$$

This implies linear growth of $h_{\nu_t}^1(\bar{G})$ for $t > t_1 + 2M/\eta$, contradicting the original assumption that $\|h_{\bar{\nu}}^1 - h_{\nu_{t_1}}^1\|_{BL} < \varepsilon$ for all $t > t_1$.

(ii) Consider now the complementary case $\nu_{t_1}(\mathcal{G}_+) = 0$. We proceed to show that there exists $t_2 > t_1$ such that $\nu_{t_2}(\mathcal{G}_+) > 0$, thereby reducing this case at time $t_2$ to part (i). To do so, we consider $\omega^* \in \text{supp}(\nu_{t_1})$ such that $\bar{\omega}^* \in \bar{G}$ is a local minimum of $g_{\bar{\nu}}$, i.e., for which $\nabla g_{\bar{\nu}} = 0$ (which exists by the preservation of the support property Assumption 1 c) ). Then, choosing $\tilde{\varepsilon}$ such that $\mathcal{B}_{\tilde{\varepsilon}}(\bar{\omega}^*) \subset \bar{G}$, and setting $M$ large enough that supp $(\nu_{t_1}) \subseteq [-M, M] \times \Theta$, we prove below in Lemma C.4 that there exists $t_2 > t_1$ for which the image at $t_2$ of $\omega(t_1) := \omega^*$ under the characteristic flow (C.6) is contained in $\mathcal{G}_+$. By continuity of the flow map $X(\cdot, t)$, this conclusion extends to a neighborhood of $\omega^*$, with positive mass under $\nu_{t_1}$.

$\square$

We denote throughout by $\|\cdot\|_{C^1}$ the maximum of the supremum norm of a function and the supremum norm of its gradient and recall the structure of the policy gradient vector field

$$F_t(\omega, \nu_t) = -\nabla\langle\omega_0\phi(\bar{\omega}), \delta(\nu_t)\rangle_{\pi_\nu} = -\nabla(\omega_0\, g_{\nu_t}(\bar{\omega})) \tag{C.10}$$

where $g$ is defined in (C.7) and $\nu_t$ solves (10).

With these definitions, we proceed to prove that case (ii) in the analysis above will ultimately reduce to case (i) for $t$ large enough.

**Lemma C.4.** *Let $\tilde{\nu} \in \mathcal{M}_+^1(\Omega)$ and $\bar{\omega}^*$ satisfy $|\nabla g_{\bar{\nu}}(\bar{\omega}^*)| = 0$, $g_{\bar{\nu}}(\bar{\omega}^*) < -\eta < 0$ for some $\eta > 0$. Then for every $\tilde{\varepsilon}, M > 0$ there exists $t_2, \varepsilon > 0$ such that if for all $t \in (0, t_2)$ we have $\|g_{\bar{\nu}} - g_{\nu_t}\|_{C^1} < \varepsilon$ and $\omega_0^* \in [-M, 0]$, then the point $\omega^*$ is mapped, under the flow of the policy gradient vector field (C.10) at time $t_2$ to a subset of $\mathcal{B}_{\tilde{\varepsilon}}((1, \bar{\omega}^*))$.*

*Proof of Lemma C.4.* By homogeneity of the approximator, we can bound the first component of the velocity of a particle $(\omega_0(t), \bar{\omega}(t))$ under (C.10) with initial condition $\omega(0) = \bar{\omega}^*$ as

$$\frac{\mathrm{d}}{\mathrm{d}t}\omega_0(t) = -g_{\nu_t}(\bar{\omega}(t)) \geq -g_{\bar{\nu}}(\bar{\omega}^*) - |g_{\bar{\nu}}(\bar{\omega}(t)) - g_{\bar{\nu}}(\bar{\omega}^*)| - |g_{\nu_t}(\bar{\omega}(t)) - g_{\bar{\nu}}(\bar{\omega}(t))|$$

In the other directions, defining $q(t) := \|\bar{\omega}(t) - \bar{\omega}^*\|$, we have

$$\frac{\mathrm{d}}{\mathrm{d}t}q(t) \leq |\omega_0(t)|\|\nabla_{\bar{\omega}}g_{\nu_t}(\bar{\omega}(t))\|$$
$$\leq |\omega_0(t)|\left[\|\nabla_{\bar{\omega}}g_{\bar{\nu}}(\bar{\omega}^*)\| + \|\nabla_{\bar{\omega}}g_{\bar{\nu}}(\bar{\omega}(t)) - \nabla_{\bar{\omega}}g_{\bar{\nu}}(\bar{\omega}^*)\| + \|\nabla_{\bar{\omega}}g_{\nu_t}(\bar{\omega}(t)) - \nabla_{\bar{\omega}}g_{\bar{\nu}}(\bar{\omega}(t))\|\right]$$

for all $t \in [0, \bar{\tau}]$ where $\bar{\tau} := \inf\{t : \omega_0(t) \notin [-M, 1]\}$. Moreover, Lipschitz continuity of the potential $g_{\bar{\nu}}(\cdot)$ and its Lipschitz smoothness imply the existence of a $L > 0$ such that $\max\{|g_{\bar{\nu}}(\bar{\omega}) - g_{\bar{\nu}}(\bar{\omega}^*)|, \|\nabla g_{\bar{\nu}}(\bar{\omega}) - \nabla g_{\bar{\nu}}(\bar{\omega}^*)\|\} \leq L\|\bar{\omega} - \bar{\omega}^*\|$. Combining this with the assumed convergence of $\nu_t$ to $\tilde{\nu}$, which implies $\|g_{\bar{\nu}} - g_{\nu_t}\|_{C^1} < \varepsilon$, we can bound the evolution of $(\omega_0(t), q(t))$ for $t \in [0, \bar{\tau}]$ in the perturbative regime of interest as follows:

$$\frac{\mathrm{d}}{\mathrm{d}t}\omega_0(t) \geq \eta - \varepsilon - Lq(t) \tag{C.11}$$

$$\frac{\mathrm{d}}{\mathrm{d}t}q(t) \leq |\omega_0(t)|\left[\varepsilon + Lq(t)\right] \tag{C.12}$$

We now show that, choosing both $\varepsilon$ and a neighborhood around $\omega^* = (\omega_0^*, \bar{\omega}^*)$ to be small enough, the forward dynamics of $\omega^*$ will reach the set $\{\omega_0 > 0\}$ *before* $q(t)$ can increase too much. More precisely, by possibly increasing the value of $L$ such that $\eta/4L < \tilde{\varepsilon}$, and defining $\tau_q = \inf\{t : q(t) > \eta/4L\}$ we prove that there

exists $\varepsilon \in (0, \eta/4)$ such that $\tau_q > \bar{\tau}$, *i.e.*, that the trajectory $\omega^*$ reaches $\mathcal{G}_+$ before $q(t) > \tilde{\varepsilon}$. Note that as long as $t \in [0, \tau_q]$ and $\varepsilon \in (0, \eta/4)$ the negative terms on the RHS (C.11) can be bounded from below, and we have

$$\omega_0(t) \geq \omega_0(0) + \frac{\eta}{2}t$$

so that $\omega_0(t) > \omega_0(0) \geq -M$. Consequently, for all $t \in [0, \bar{\tau} \wedge \tau_q]$ we bound the RHS of (C.12) as $\frac{\mathrm{d}}{\mathrm{d}t}q(t) < M\varepsilon + LMq(t)$. Using that $q(0) = 0$ and Grönwall inequality we can bound the total excursion in the $\bar{\omega}$ component $q(t) \leq \varepsilon M t \exp[LMt]$. Finally, setting $\tau_0 := 2(M+1)/\eta \geq -2(\omega_0(0) - 1)/\eta > \bar{\tau}$ so that $\omega_0(\tau_0) > 1$ we are still free to set $\varepsilon$ small enough such that $\tau_q > \tau_0 > \bar{\tau}$. Indeed, by monotonicity of the upper-bound on $q(t)$ we have

$$q(\tau_0) \leq 2\varepsilon(M+1)M/\eta \exp[2LM(M+1))/\eta] \leq \eta/4L,$$

so that setting $\varepsilon \in (0, \eta/4)$ concludes the proof. $\qquad\square$

We now proceed to show the needed regularity of the potential $g_\nu$ from (C.7) in terms of the signed measure $h^1_\nu$ defined in (C.5). In doing so, we also prove Lipschitz smoothness of the operator $R$ defined in (B.1):

**Lemma C.5.** *For any $r > 0$, the operator $R$ on $\mathcal{F}_r = \{\int \psi \nu : \operatorname{supp}\nu \in \mathcal{Q}_r\}$ is Lipschitz smooth, and $DR_f$ is bounded in the supremum norm. Furthermore, for all $C_0 > 0$ there exists $\alpha > 0$ and $\varepsilon > 0$ such that for all $\nu, \nu'$ satisfying $\|h^1_\nu\|_{BL}, \|h^1_{\nu'}\|_{BL} < C_0$, $\|h^1_\nu - h^1_{\nu'}\|_{BL} < \varepsilon$, one has*

$$\|g_\nu - g_{\nu'}\|_{C^1} \leq \alpha\|h^1_\nu - h^1_{\nu'}\|_{BL}. \tag{C.13}$$

To prove the above lemma, we first bound some relevant quantities. Throughout, by slight abuse of notation, we denote for any function $f : \mathcal{S} \times \mathcal{A} \to \mathcal{R}$

$$\|f\|_2 = \sup_{s \in \mathcal{S}} \int_{\mathcal{A}} f(s, a)^2 \mathrm{d}a$$

**Lemma C.6.** *For all $f, f' \in \mathcal{F}_r$ there exists $\varepsilon > 0$, $C', C'', C''' > 0$ such that if $\|f - f'\|_2 < \varepsilon$ one has*

$$\|\pi_\nu - \pi_{\nu'}\|_2 \leq C'\|f - f'\|_2 \tag{C.14}$$
$$\|\varrho_\nu - \varrho_{\nu'}\|_1 \leq C''\|f - f'\|_2 \tag{C.15}$$
$$\|Q_{\pi_\nu} - Q_{\pi_{\nu'}}\|_2 \leq C'''\|f - f'\|_2 \tag{C.16}$$

*Proof of Lemma C.6.* Throughout this proof, for simplicity of notation, we will write $\pi = \pi_\nu$ and $\pi' = \pi_{\nu'}$. Furthermore, we use that for $f \in \mathcal{F}_r$ there exists $C_0 > 0$ so that

$$e^{-C_0\|\phi\|_{C^1}} \leq \exp[f(s, a)] \leq e^{C_0\|\phi\|_{C^1}}, \tag{C.17}$$

implying, together with the assumed absolute continuity of $\varrho_0$ that $\|Q_\pi\|_\infty, \|\pi\|_\infty, \|\varrho_\pi\|_\infty < \infty$. Setting throughout $\tau = 1$ to simplify the notation and combining the above with the pointwise upper bound $e^x < 1 + K_r|x|$ for $|x| < e^{C_0\|\phi\|_{C^1}}$ we obtain

$$
\begin{aligned}
\|\pi - \pi'\|_2 &\leq \left\| \frac{\exp[f(s, a)]}{\int \exp[f(s, a')]\mathrm{d}a'} - \frac{\exp[f'(s, a)]}{\int \exp[f'(s, a')]\mathrm{d}a'} \right\|_2 \\
&\leq \left\| \frac{\exp[f(s, a)]}{\int \exp[f(s, a')]\mathrm{d}a'} - \frac{\exp[f'(s, a)]}{\int \exp[f(s, a')]\mathrm{d}a'} \right\|_2 + \left\| \frac{\exp[f'(s, a)]}{\int \exp[f(s, a')]\mathrm{d}a'} - \frac{\exp[f'(s, a)]}{\int \exp[f'(s, a')]\mathrm{d}a'} \right\|_2 \\
&\leq \left\| \frac{\exp[f'(s, a)]}{\int \exp[f(s, a')]\mathrm{d}a'} \right\|_\infty \left( \|1 - \exp[f(s, a) - f'(s, a)]\|_2 + \left\| \frac{\int \exp[f(s, a')]\mathrm{d}a'}{\int \exp[f'(s, a')]\mathrm{d}a'} - 1 \right\|_2 \right) \\
&\leq \frac{e^{2C_0\|\phi\|_{C^1}}}{|\mathcal{A}|} \left( K_r \|f(s, a) - f'(s, a)\|_2 + \frac{e^{2C_0\|\phi\|_{C^1}}}{|\mathcal{A}|} \left\| \int (\exp[f(s, a') - f'(s, a')] - 1)\mathrm{d}a' \right\|_2 \right) \\
&\leq \frac{e^{2C_0\|\phi\|_{C^1}}}{|\mathcal{A}|} K_r(1 + e^{4C_0\|\phi\|_{C^1}})\|f - f'\|_2 =: C'\|f - f'\|_2, \tag{C.18}
\end{aligned}
$$

where we have denoted by $|\mathcal{A}|$ the Lebesgue measure of the action space $\mathcal{A}$. We now proceed to establish the second bound in the statement of the lemma. In this case, denoting the $t$-steps transition probability as $P_\pi^t(s, \mathrm{d}s_t) = \int_{\mathcal{S}^{t-1}} P_\pi(s, \mathrm{d}s_1) P_\pi(s_1, \mathrm{d}s_2) \ldots P_\pi(s_{t-1}, \mathrm{d}s_t)$ we have

$$\|\varrho_\pi - \varrho_{\pi'}\|_1 = \left\| \sum_{t=1}^{\infty} \gamma^t \int_{\mathcal{S}} \varrho_0(\mathrm{d}s_0) \left( P_\pi^t(s_0, \mathrm{d}s) - P_{\pi'}^t(s_0, \mathrm{d}s) \right) \right\|_1$$

$$\leq \sum_{t=1}^{\infty} \gamma^t \sum_{j=0}^{t-1} \left\| \varrho_0 P_\pi^j \left( P_\pi - P_{\pi'} \right) P_{\pi'}^{t-j-1} \right\|_1 .$$

Observing that for any smooth $\varrho \in \mathcal{M}_+^1(\mathcal{S})$ for the operator norm of the difference in the above sum we have

$$\|\varrho(P_\pi - P_{\pi'})\|_1 = \left\| \int_{\mathcal{S}} \varrho(\mathrm{d}s) \int_{\mathcal{A}} P(s, a, \mathrm{d}s') \left( \pi(s, \mathrm{d}a) - \pi'(s, \mathrm{d}a) \right) \right\|_1$$

$$\leq \|\varrho\|_1 \|P\|_1 \|\pi - \pi'\|_2 \leq \frac{(1-\gamma)^2}{\gamma} C'' \|f - f'\|_2$$

for large enough $C''$, where we used (C.18) in the last line and the Lipschitz continuity of $P$ in its second argument, and $\|P\|_1$ is the operator norm of the transition operator $\int_{\mathcal{A}} P(s, a, \mathrm{d}s') \pi''(s, \mathrm{d}a) : \mathcal{M}_+^1(\mathcal{A}) \to \mathcal{M}_+^1(\mathcal{A})$, which is equal to 1. From this we conclude

$$\|\varrho_\pi - \varrho_{\pi'}\|_1 \leq \frac{(1-\gamma)^2}{\gamma} \sum_{t=1}^{\infty} t \gamma^t C'' \|f - f'\|_2 = C'' \|f - f'\|_2$$

Finally, defining for notational convenience $R'_\tau := \bar{r} - \tau D_{KL}(\pi', \bar{\pi})$ we write:

$$\|Q_\pi - Q_{\pi'}\|_2 = \left\| \gamma \int P(s, a, \mathrm{d}s') \left( V_\pi(s') - V_{\pi'}(s') \right) \right\|_2$$

$$= \left\| \gamma \int P(s, a, \mathrm{d}s') \left( \int R_\tau(s'', a') \varrho^\pi(s', \mathrm{d}s'') \pi(\mathrm{d}a') - R'_\tau(s'', a') \varrho^{\pi'}(s', \mathrm{d}s'') \pi'(\mathrm{d}a') \right) \right\|_2$$

$$\leq \gamma \left\| \int P(s, a, \mathrm{d}s') \left( R_\tau(s'', a') \varrho^\pi(s', \mathrm{d}s'') \pi(\mathrm{d}a') - R'_\tau(s'', a') \varrho^{\pi'}(s', \mathrm{d}s'') \pi'(\mathrm{d}a') \right) \right\|_2$$

$$\leq \left\| \int (R_\tau - R'_\tau)(s'', a') \varrho^\pi(s', \mathrm{d}s'') \pi(\mathrm{d}a') \right\|_\infty + \left\| \int R'_\tau(s'', a')(\varrho^\pi - \varrho^{\pi'})(s', \mathrm{d}s'') \pi(\mathrm{d}a') \right\|_\infty$$

$$+ \left\| \int R'_\tau(s'', a') \varrho^{\pi'}(s', \mathrm{d}s'')(\pi - \pi')(\mathrm{d}a') \right\|_\infty \tag{C.19}$$

and bound each term separately letting $C_1''', C_2''', C_3''' > 0$ be large enough constants. For the first, we have:

$$\left\| \int (R_\tau - R'_\tau)(s'', a') \varrho_\pi(s', \mathrm{d}s'') \pi(s'', \mathrm{d}a') \right\|_\infty \leq \frac{1}{1-\gamma} \|\pi\|_2 \|\log \pi - \log \pi'\|_2$$

$$\leq \frac{1}{1-\gamma} \|\pi\|_2 \left( \|f - f'\|_2 + \left\| \log \frac{\int e^{f(s,a)} \mathrm{d}a}{\int e^{f'(s,a)} \mathrm{d}a} \right\|_2 \right)$$

$$\leq C_1''' \|f - f'\|_2 \tag{C.20}$$

where we have bounded the log term as follows:

$$\left\| \log \frac{\int e^{f(s,a)} \mathrm{d}a}{\int e^{f'(s,a)} \mathrm{d}a} \right\|_2 = \left\| \log \frac{\int e^{f(s,a)} - e^{f'(s,a)} \mathrm{d}a}{\int e^{f'(s,a)} \mathrm{d}a} + 1 \right\|_2$$

$$\leq \left\| \log \frac{\int e^{f'(s,a)} \left( e^{|f(s,a) - f'(s,a)|} - 1 \right) \mathrm{d}a}{\int e^{f'(s,a)} \mathrm{d}a} + 1 \right\|_2$$

$$\leq \| \log \left( \frac{\|e^{-f'}\|_\infty \|e^{f'}\|_\infty}{|\mathcal{A}|} \int \left( e^{|f(s,a)-f'(s,a)|} - 1 \right) \mathrm{d}a + 1 \right) \|_2$$

$$\leq \| \log \left( \frac{\|e^{-f'}\|_\infty \|e^{f'}\|_\infty}{|\mathcal{A}|} K_r \int |f(s,a) - f'(s,a)| \mathrm{d}a + 1 \right) \|_2$$

$$\leq \| \log \left( \|e^{-f'}\|_\infty \|e^{f'}\|_\infty K_r \|f(s,a) - f'(s,a)\|_2 + 1 \right) \|_2$$

$$\leq \|e^{-f'}\|_\infty \|e^{f'}\|_\infty K_r \|f - f'\|_2 \tag{C.21}$$

For the second term in (C.19), using the boundedness of $\|R\|_2 < C_R$ and that $\varrho^\pi - \varrho^{\pi'} = \gamma (\varrho_\pi - \varrho_{\pi'})$ for a certain, $\varrho_0$ depending on $s'$ we write

$$\left\| \int R'_\tau(s'',a')(\varrho^\pi - \varrho^{\pi'})(s',\mathrm{d}s'')\pi(s'',\mathrm{d}a') \right\|_\infty \leq \|\pi\|_2 \|R'_\tau(s'',a')\|_2 \|\varrho_\pi - \varrho_{\pi'}\|_1$$

$$\leq C_2''' \|f - f'\|_2. \tag{C.22}$$

We finally bound the third term by writing

$$\left\| \int R'_\tau(s'',a')\varrho_{\pi'}(s',\mathrm{d}s'')(\pi - \pi')(s'',\mathrm{d}a') \right\|_\infty \leq \|\varrho_{\pi'}\|_1 \|R'_\tau(s'',a')\|_2 \|\pi - \pi'\|_2$$

$$\leq C_3''' \|f - f'\|_2 \tag{C.23}$$

and obtain (C.16) by combining (C.20)-(C.23). $\qquad\square$

*Proof of Lemma C.5.* We first establish the desired properties of the functional $R$. To do so we differentiate (8) at $f \in L^2(\mathcal{S} \times \mathcal{A})$

$$\frac{\delta \pi_f}{\delta f}(s,a) = \frac{\delta}{\delta f} \frac{e^{\tau \int f(s,a)\nu(\mathrm{d}\omega)}}{\int_\mathcal{A} e^{\tau f(s,a)} \mathrm{d}a} = \frac{1}{\int_\mathcal{A} e^{\tau f} \mathrm{d}a} \frac{\delta}{\delta f} e^{\tau f} - \frac{e^{\tau f}}{\left( \int_\mathcal{A} e^{\tau f} \mathrm{d}a \right)^2} \int_\mathcal{A} \frac{\delta}{\delta \nu} e^{\tau f} \mathrm{d}a$$

$$= \tau \left( f - \int_\mathcal{A} f \pi_f(s,\mathrm{d}a') \right) \pi_f(s,a). \tag{C.24}$$

Then, combining the above with (A.6) and Hölder inequality we obtain

$$\|DR_f\|_{\infty,r} = \sup_{f \in \mathcal{F}_r} \int (DR_f(s,a))^2 \mathrm{d}s\mathrm{d}a < \infty$$

where we have used that all the terms appearing in $DR_f$ are bounded for every choice of $r > 0$.

To establish Lipschitz smoothness of $R$, letting $f, f' \in \mathcal{F}_r$ and denoting, to simplify notation, $\pi = \pi_f$ and $\pi' = \pi_{f'}$, we proceed to bound the operator norm by splitting the RHS as

$$\|DR_f - DR_{f'}\| \leq \sup_{\ell \in L^2(\mathcal{S} \times \mathcal{A}) \, : \, \|\ell\|=1} |C_\pi[\ell, Q_\pi - \tau \log \pi] - C_{\pi'}[\ell, Q_{\pi'} - \tau \log \pi']|$$

$$\leq \sup_{\ell \in L^2(\mathcal{S} \times \mathcal{A}) \, : \, \|\ell\|=1} [(I) + (II) + (III)]$$

and considering the resulting terms separately.

First of all, defining throughout by slight abuse of notation $\delta(\pi) := Q_\pi - \tau \log \pi$, we have

$$(I) := |\langle \int \ell(s,a)\pi'(s,\mathrm{d}a) - \int \ell(s,a)\pi(s,\mathrm{d}a), \delta(\pi')\rangle_{\pi'}|$$

$$\leq \int_\mathcal{S} \left( \int_\mathcal{A} \ell(s,a)(\pi(s,a) - \pi'(s,a))\mathrm{d}a \right) \left( \int \delta(\pi')\pi'(\mathrm{d}a) \right) \varrho_{\pi'}(s)\mathrm{d}s$$

$$\leq \|\ell\|_2 \|\pi - \pi'\|_2 \|\delta(\pi')\pi'\|_2 \|\varrho_{\pi'}\|_1$$

$$\leq C \|\ell\|_2 \|f - f'\|_2, \tag{C.25}$$

where the last line was obtained using (C.14) and boundedness from above and below of $\pi, Q_\pi$ in $\mathcal{F}_r$. We further write, for a $K < \infty$ large enough

$$
\begin{aligned}
(II) &:= |\langle \ell(s,a) - \int \ell(s,a')\pi(s,\mathrm{d}a'), \delta(\pi') \rangle_{\pi'} - \langle \ell(s,a) - \int \ell(s,a')\pi(s,\mathrm{d}a'), \delta(\pi') \rangle_\pi| \\
&\leq |\int \delta(\pi')\ell(s,a)\varrho_\pi(s)(\pi - \pi')(s,\mathrm{d}a)\mathrm{d}s| + |\int \ell(s,a)\delta(\pi')(\varrho_\pi - \varrho_{\pi'})(s)\pi'(s,\mathrm{d}a)\mathrm{d}s| \\
&\leq \|\delta(\pi')\|_2 \|\varrho_{\pi'}\|_1 \|\ell\|_2 \left( \|\pi - \pi'\|_2 + \|\varrho_\pi - \varrho_{\pi'}\|_1 \right) \\
&\leq K \|\ell\|_2 \|f - f'\|_2,
\end{aligned} \tag{C.26}
$$

where we have used Cauchy-Schwartz inequality together with (C.14) and (C.15). Finally, using (C.16) and (C.21), we bound for $K < \infty$ possibly larger than above

$$
\begin{aligned}
(III) &:= |\langle \ell(s,a) - \int \ell(s,a')\pi(s,\mathrm{d}a'), \delta(\pi) - \delta(\pi') \rangle_\pi| \\
&\leq \|\varrho_\pi \pi\|_\infty \|\ell\|_2 \|\delta(\pi) - \delta(\pi')\|_2 \\
&\leq \|\varrho_\pi\|_1 \|\pi\|_2 \|\ell\|_2 \left( \|Q_\pi - Q_{\pi'}\|_2 + \|V_\pi - V_{\pi'}\|_2 + \tau \|\log \frac{\pi}{\pi'}\|_2 \right) \\
&\leq \|\varrho_\pi\|_1 \|\pi\|_2 \|\ell\|_2 \left( (1 + \|\pi'\|_\infty)\|Q_\pi - Q_{\pi'}\|_2 + \|Q_\pi\|_2 \|\pi - \pi'\|_2 + \tau \|\log \frac{\pi}{\pi'}\|_2 \right) \\
&\leq K \|\ell\|_2 \|f - f'\|_2.
\end{aligned} \tag{C.27}
$$

Combining (C.25), (C.26) and (C.27) we obtain that

$$
L_{DR} = \sup_{f,f' \in \mathcal{F}_r} \frac{\|DR_f - DR_{f'}\|}{\|f - f'\|_2} < \infty
$$

proving the Lipschitz smoothness claim.

Proceeding to the proof of (C.13), combining the Lipschitz smoothness of $R$ on bounded sets, the identity $\psi(s,a;(1,\bar{\omega})) = \phi(s,a;\bar{\omega})$ and the boundedness of the set $\{\int \psi \nu \ : \ \nu \in \mathcal{P}_2(\Omega), |h_\nu^1| < C_0\}$ we obtain

$$
\begin{aligned}
\|f_\nu - f_{\nu'}\|_2 &= \left\| \int \psi(\cdot;\omega)(\nu - \nu')(\mathrm{d}\omega) \right\|_2 \tag{C.28} \\
&= \left\| \int \phi(\cdot;\omega)(h_\nu^1 - h_{\nu'}^1)(\mathrm{d}\bar{\omega}) \right\|_2 \\
&\leq \sup_{\ell \in L^2(\mathcal{S} \times \mathcal{A}), \|\ell\| \leq 1} \int \int \ell(s,a)\phi(s,a;\bar{\omega})\mathrm{d}s\mathrm{d}a(h_\nu^1 - h_{\nu'}^1)(\mathrm{d}\bar{\omega}) \\
&\leq \|\phi\|_{C^1} \|h_\nu^1 - h_{\nu'}^1\|_{BL},
\end{aligned}
$$

which combined with the $\|\phi\|_{C^1}$-Lipschitz continuity of the map $\int \ell(s,a)\phi(s,a;\bar{\omega})\mathrm{d}s\mathrm{d}a$ and with the Lipschitz smoothness of $R$ concludes the proof. $\qquad \square$

