# OpenReview forum: "Global optimality of softmax policy gradient with single hidden layer neural networks in the mean-field regime"
_ICLR.cc/2021/Conference — ICLR 2021 Poster_

### Official Review · AnonReviewer1 · 2020-10-27
**Solid contribution to RL theory**

**Rating:** 7
**Confidence:** 3

**Review:**

This paper studies the mean-field limit of the policy gradient method (with entropy regularized) and proves that any stationary point under this setting is a global minimizer. I am not able to verify the entire proof as it involves a lot of standard steps to bridging the finite parameter case and the mean-field limit. The result seems promising and well complements several theory results in RL in the past year, e.g. the optimality of policy gradient under NTK regime and the TD algorithm in the mean-field regime.

Although the paper does not provide the convergence guarantee of the mean-field density flow to a stationary point (please correct me if this is wrong), the characterization of the optimality is still a good contribution. It well explains why a neural network policy is globally optimal given (1)it is stationary under the training via the first-order method (2)its parameterization has strong expressive power, e.g. it has infinite parameters or it is essentially a nonparametric model.

The convergence (section 4.2) to the many-particle limit. i.e. mean-field limit, seems standard, as the authors claim it is very similar to the case of supervised learning. I still would like to ask whether the authors found any key differences between the supervised learning case and RL objective, i.e., maximizing the total reward. In particular, does the absence of a strongly convex loss function cause any difficulty in the proof?

---

> ### Author Response · Authors · 2020-11-14
> **Response to AnonReviewer1**
>
> We thank the referee for the careful review and the encouraging remarks on our paper.
>
> We gladly address the point raised by the referee about the difference of the objective and of the convergence argument between the supervised and reinforcement learning setting.
>
> In our proof, we have found that the RL setting adds a substantial degree of nonlinearity and nonconvexity of the objective function. This results on one hand from the nonconvexity of the map $\pi \mapsto \mathbb E_\rho[V_\pi]$ - resulting from the complex dependency of $V_\pi$ on $\pi$ - and on the other from the softmax map $f \to \pi_f$, which is also nonconvex. While the latter results from a choice of the authors, the former is, a priori, intrinsic to the problem at hand: the value function is known to enjoy only one-point convexity properties even in the tabular setting for general problems. Consequently, combining this nonconvex landscape with a low expressivity of the model may result in local minima of the dynamics. We avoid this problem by requiring sufficient expressive power of the used model, which in turn allows us to take advantage of the properties of the landscape at hand. In the supervised setting, the expressivity assumption was not needed as the loss functional was assumed to be convex, thereby removing the possibility of observing spurious minima.
>
> Concerning the convergence of the particle system to the mean-fiend PDE, the main difficulty compared with the supervised setting results is to establish needed Lipschitz continuity properties of the vector field driving the transport PDE. Again, while these properties are assumed or immediate in the supervised case, proving them requires more effort in the RL setting given the more involved dependence of the vector field on the measure $\nu_t$. Once this result is established, a standard Gronwall argument concludes the proof of convergence.
>
> We stress the above point in the updated version of the paper (edits in red). Hoping that our response and revision have satisfactorily addressed the referee’s remarks, we thank them again for their insightful comments.

---

### Official Review · AnonReviewer3 · 2020-10-28
**This paper provides a mean-field formulation of policy gradient dynamics in parameter space in the reinforcement learning framework. It proves the convergence of the particle dynamics to their mean-field counterpart and the convergence of the mean-field dynamics.**

**Rating:** 7
**Confidence:** 4

**Review:**

Overall, I vote for accepting. This paper extends previous work in the parameter dynamics of simple neural networks to reinforcement learning framework in continuous state and action spaces with nonlinear function approximation and overcomes the challenge of lack of convexity. The main concern of mine would be that the theorems in the paper are not powerful enough to help us fully understand the experiments.

Pros:

1. This paper introduces the mean-field formulation into the reinforcement learning framework. The technical proof seems highly challenging.
2. Under mild conditions, it demonstrates interesting convergence properties of the particle dynamics to the mean-field counterpart and further, the mean-field dynamics to the global optima. This provides new insight into theoretical understanding of this problem.

Cons:
1. There still exists gap between the theoretical results and numerical experiments to be filled. The experiments shown in Figure 1 are conducted with finite number of neutrons and constant step-size, but the theorems are stated under the adiabatic limit. Hence there is lack of quantitative results under the setting of finite number of samples and finite gradient step size.
2. The paper only shows convergence, but fails to give more detailed properties like convergence rate, etc.

---

> ### Author Response · Authors · 2020-11-14
> **Response to AnonReviewer3**
>
> We are grateful for the positive review and for the appreciative remarks of the referee. We agree with them that there still exists a gap that needs to be filled between theory and practice, and in particular between the behavior of the finite-step and finite-neuron model and the asymptotic limit studied in this paper. While we anticipate that the error resulting from finite step-size can be controlled, over finite time intervals, by standard stochastic approximation results, the finite-width analysis may pose more complex challenges. Indeed, in the finite-width regime the full-support assumption at initialization will fail, and to establish similar optimality results as those presented in this paper, precise estimates on the empirical measure of parameters throughout the dynamics will be needed. Therefore, while we look at this question with great interest, it is beyond the scope of our current work.
>
> Similarly, the question of quantitative convergence estimates for the nonlinear dynamics at hand is, in the general case, widely open. The main difficulty in this setting results from the non-geodesic convexity of the energy of the PDE problem (with respect to the Wasserstein metric). As indicated in the response to AnonReviewer2, this issue has been bypassed in a few special cases in the recent literature, either by considering a subclass of the models [Javanmard, Mondelli, Montanari, 2019], or by regularizing the objective function [Chizat, 2019].
>
> We thank again the referee for their remarks and hope that our response could address their concerns in a satisfactory way.

---

### Official Review · AnonReviewer4 · 2020-10-29
**Review for "Global optimality of softmax policy gradient with single hidden layer neural networks in the mean-field regime"**

**Rating:** 7
**Confidence:** 3

**Review:**

This paper provides a mean-field characterization of entropy-regularized policy gradient dynamics for wide single hidden layer neural networks. The evolution of neural network parameters is described by a transport partial differential equation. And the convergence properties of the dynamics are established.

Overall, I vote for accepting. The paper is well-written. The technical contents seem sound and a comprehensive literature review is provided. And authors also conduct numerical experiments to validate the theory. My two minor comments are as follows:

- The remarks in Section 4.1 are mainly explaining why authors need the assumptions to establish the theories. It would be nicer to provide examples when these assumptions hold or discuss the generality of the assumptions.

- It would be nicer to spend more space to discuss the major differences of the theoretical analysis in this paper compared to earlier results on the mean-field limit in the supervised learning setting. It would be helpful to discuss different aspects of the technical analysis more explicitly.

---

> ### Author Response · Authors · 2020-11-14
> **Response to AnonReviewer4**
>
> We thank the referee for the positive review. We address their helpful comments below. We have revised the paper according to the suggestions, which hopefully improve the presentation of our results.
>
> For each assumption in Section 4.1, we have included some examples and comments in the revised version of the paper, which we also list here:
> 1. The conditions of boundedness and continuity of the nonlinearity $\phi$ are for instance satisfied by logistic, tanh and Gaussian radial function nonlinearities. Further extension to the ReLU case was discussed in [Wojtowytsch, 2020] for supervised learning,
> 2. The expressivity condition on the nonlinearity $\phi$ is for instance satisfied by ReLU, logistic, tanh and gaussian radial function nonlinearities,
> 3. The full support assumption on the measure at initialization holds for instance for the product of a uniform distribution on any bounded set $A \subset \mathbb R$ (for the last layer) with the normal distribution on $\Theta$ or, if $\Theta$ is compact, with the uniform distribution on $\Theta$ (for the first layer).
>
> We have also revised the section where we sketch the proof of the main result of the paper, further  stressing the difference between the RL analysis and its supervised learning counterpart (edits in red).
>
> We hope that our clarifications could exhaustively address the referee’s comments.

---

### Official Review · AnonReviewer2 · 2020-10-30
**Interesting asymptotic results on policy gradient methods**

**Rating:** 7
**Confidence:** 3

**Review:**

This paper studies the asymptotic convergence properties of (population-level) policy gradient methods with two-layer neural networks, softmax parametrization, and entropic regularization, in the mean-field regime. By modelling the hidden layer as a probability distribution over the parameter space, the training dynamics of policy gradient methods can be written as a partial differential equation. Under certain regularity conditions, the paper shows that if the training dynamics converge to a stationary point, this limiting point is a globally optimal policy. The paper also presents results for finite-time convergence of the training dynamics for neural networks to the mean-field limit.

The optimization landscape and convergence properties of policy gradient methods have drawn attention in RL theory for a long time, and it is nice to see a work that studies this problem from the perspectives of mean-field limit of neural networks, albeit being completely asymptotic. Overall I think this makes an interesting contribution, and I appreciate the sketch of proof ideas in the simpler bandit case. Technically, it seems that the main results are built upon existing frameworks of Mei et al., (2018), Chizat and Bach et al., (2018), etc. But the author also pointed out an interesting technical novelty, which is the use of density arguments when the problem structure is in lack of the hidden convexity used in other works.

On the other hand, it appears to me that one major weakness of the result is that the theorem holds true only when the dynamics converges to a stationary point. Can the authors provide conditions under which this can happen? For example, would it be possible to establish some compactness under additional regularity conditions and use it to show the convergence of a subsequence? If the convergence does fail to happen in certain regimes, how will the dynamics behave? Will it convergence to a limiting cycle or diverge? Are there some natural counter-examples? It would be helpful if the authors could provide more discussions on this condition.

Additionally, it seems to me that the paper actually shows that (due to entropic regularization) the limiting point is the Boltzman policy induced by the optimal Q function (at temparature $\tau$), instead of the optimal Q function iteslf. If that is the case, this needs to be stated clearly in the theorem.

---

> ### Author Response · Authors · 2020-11-14
> **Response to AnonReviewer2**
>
> We thank the referee for the positive review and address their questions below.
>
> For the question on convergence, while some recent papers have obtained convergence guarantees for mean-field approximators in the supervised setting [Javanmard, Mondelli, Montanari, 2019], [Chizat, 2019], [Mei, Montanari, Nuyen, 2018], in the fully general setting the question of convergence of mean-field models remains widely open. More specifically, in [Javanmard, Mondelli, Montanari, 2019] the family of approximators being considered fixes the weights of the last layer, obtaining convergence results by the resulting convexity of the landscape in Wasserstein space, while [Chizat, 2019], [Mei, Montanari, Nuyen, 2018] obtain convergence properties of the model by regularizing the loss function (explicitly or implicitly). These works give examples of how, under further regularity assumptions, it is indeed possible to establish compactness and convergence of the model. We expect similar results to hold in the RL setting, at the possible expense of a certain degree of suboptimality of the limiting policy, depending on the compatibility between the regularization and the “planted” policy (for instance, if the optimal policy is given by a finite-width neural network, then one would expect adding sparsifying regularization to enable policy gradient to find such a policy).
> As for the general case, if the convergence fails to happen it is not clear, in general, what kind of non-convergent behaviour will be displayed by the dynamics. One important point of distinction in this sense is whether the optimal approximator lies in the space of functions that can be approximated by the neural network with the given nonlinearity: if this is not the case divergent behavior has been observed in [Chizat, Bach, 2020] while, to the best of the authors’ knowledge, no limit cycle dynamics in parameter space has so far appeared in the literature. If, on the other hand, the target approximator does lie in the approximation space of the model at hand, to the best of our knowledge no counter-examples to the conjectured convergent behavior are known.
>
> For the limiting policy, we agree with the referee that the optimal policy is indeed the Boltzmann policy induced by the optimal  Q function for the entropy-regularized problem, and not the one induced by the optimal Q function of the underlying, unregularized problem. We further clarify this point in the updated version of the manuscript (edits in red).
>
> We hope that our response and revision have satisfactorily addressed the referee’s questions.

---

### Author Response · Authors · 2020-11-24
**Final comment for reviewers**

We thank again the reviewers for their interesting questions and insightful comments, according to which we have updated our submission as discussed in the personalized replies below. We have uploaded a final version removing the red parts indicating changes wrt the original submission. We have also made some minor changes in to improve presentation and readibility.

---

### Decision · Program_Chairs · 2021-01-07
**Final Decision**

**Decision:**

Accept (Poster)

**Comment:**

This paper takes a step towards understanding the role of nonlinear function approximation--- more specifically, function approximation via (two-layer) neural nets---in some variants of the policy-gradient algorithms. The authors borrow the mean field analysis idea recently popularized in studying shallow neural nets, and investigate the mean-field limits of the training dynamics in the current RL settings. The results and analyses are interesting as they nicely complement another line of linearization-based analyses (i.e., the one based on neural tangent kernels) towards understanding non-linear function approximation. As suggested by a reviewer, it would be nice to add discussions in the revised paper regarding when the dynamics can be guaranteed to converge to a stationary point.